# Endogenous FGF21-signaling controls paradoxical obesity resistance of UCP1-deficient mice

Susanne Keipert [1,2,3]*, Dominik Lutter [1,2], Bjoern O. Schroeder [4,9], Daniel Brandt[1,2], Marcus Ståhlman[4], Thomas Schwarzmayr[5], Elisabeth Graf[5], Helmut Fuchs [6], Martin Hrabe de Angelis [2,6,7], Matthias H. Tschöp [1,2,8], Jan Rozman [2,6,10] & Martin Jastroch[1,2,3]*

Uncoupling protein 1 (UCP1) executes thermogenesis in brown adipose tissue, which is a major focus of human obesity research. Although the UCP1-knockout (UCP1 KO) mouse represents the most frequently applied animal model to judge the anti-obesity effects of UCP1, the assessment is confounded by unknown anti-obesity factors causing paradoxical obesity resistance below thermoneutral temperatures. Here we identify the enigmatic factor as endogenous FGF21, which is primarily mediating obesity resistance. The generation of UCP1/FGF21 double-knockout mice (dKO) fully reverses obesity resistance. Within mild differences in energy metabolism, urine metabolomics uncover increased secretion of acyl-carnitines in UCP1 KOs, suggesting metabolic reprogramming. Strikingly, transcriptomics of metabolically important organs reveal enhanced lipid and oxidative metabolism in specifically white adipose tissue that is fully reversed in dKO mice. Collectively, this study characterizes the effects of endogenous FGF21 that acts as master regulator to protect from diet-induced obesity in the absence of UCP1.

[1] Institute for Diabetes and Obesity, Helmholtz Diabetes Center, Helmholtz Zentrum München, German Research Center for Environmental Health (GmbH), 85764 Neuherberg, Germany. [2] German Center for Diabetes Research (DZD), 85764 Neuherberg, Germany. [3] Department of Molecular Biosciences, The Wenner-Gren Institute, Stockholm University, 106 91 Stockholm, Sweden. [4] Wallenberg Laboratory and Sahlgrenska Center for Cardiovascular and Metabolic Research, Institute of Medicine, University of Gothenburg, 413 45 Göteborg, Sweden. [5] Institute of Human Genetics, Helmholtz Zentrum München, German Research Center for Environmental Health (GmbH), 85764 Neuherberg, Germany. [6] German Mouse Clinic, Institute of Experimental Genetics, Helmholtz Zentrum München, German Research Center for Environmental Health (GmbH), 85764 Neuherberg, Germany. [7] Chair of Experimental Genetics, School of Life Science, Weihenstephan, Technische Universität München, 85354 Freising, Germany. [8] Division of Metabolic Diseases, Department of Medicine, Technische Universität, Munich, Germany. [9] Present address: Laboratory for Molecular Infection Medicine Sweden (MIMS) and Department of Molecular Biology, Umeå University, 901 87 Umeå, Sweden. [10] Present address: Czech Centre for Phenogenomics, Institute of Molecular Genetics of the Czech Academy of Sciences BIOCEV, 252 50 Vestec, Czech Republic. *email: Susanne.keipert@su.se; martin.jastroch@su.se

Obesity is a major health burden of our society that causes comorbidities, such as diabetes, heart disease, and cancer. Positive energy balance triggers obesity, either by increased food intake and high-calorie diets, or by decreased energy dissipation, or a combination of both[1,2]. While food intake and preference are mainly regulated via central mechanisms, energy dissipation takes place in peripheral organs. Thermogenic adipose tissue is considered an attractive target for the treatment of obesity and diabetes, evidenced by numerous recent studies that aim to activate energy dissipation by increasing heat production[3,4]. Rodent models established that defective brown adipose tissue (BAT) thermogenesis is involved in the development of obesity, and activation of BAT thermogenesis reduces weight gain[5–9]. Albeit the presence of BAT in adult humans is well-accepted[10–13], its contribution to whole body metabolism is a matter of debate[14]. Neonatal BAT mass is limited in adults, while a larger proportion has been classified as beige adipose tissue[15,16]. The physiological role of beige adipose tissue has not been clarified yet, but could be the target tissue for therapeutic intervention in humans. In mice, beige fat mass increases when BAT mass is genetically reduced, suggesting compensatory heat production[5]. The subcutaneous beige fat, however, may not be able to distribute heat as effectively as the neonatal BAT, that directly delivers heat to the heart via the Sulzer's vein. From the therapeutic perspective, heat generation in the subcutaneous part of the body could be advantageous as increased heat dissipation may prevent hyperthermia in humans.

Thermogenic adipose-specific mitochondrial uncoupling protein 1 (UCP1) is solely responsible for the canonical mechanism of adaptive heat production[17] and is expressed in both, brown and beige fat[18]. Activation of UCP1 increases the mitochondrial proton leak, thereby uncoupling mitochondrial respiration from ATP synthesis[19]. The acceleration of nutrient oxidation increases all exergonic reactions, which results in increased heat output. Notably, obese humans express only minor amounts of UCP1; and thus, UCP1-independent thermogenic mechanisms in adipose tissue may bear higher translational potential[20]. Interestingly, the importance of UCP1-dependent thermogenesis for body weight regulation is mainly supported by a proof-of-principle study of UCP1-knockout (KO) mice at thermoneutrality, which are slightly prone to diet-induced obesity (DIO)[9]. Paradoxically, UCP1 KO mice are surprisingly resistant to DIO at standard mouse housing temperatures of 20–24 °C[21–23]. Neither the regulation nor the underlying molecular mechanisms have been understood yet, labeling the observation as 'paradoxical resistance to diet-induced obesity of high fat diet fed UCP1 KO mice'[23]. A comprehensive compendium on BAT biology stated that there must be secretion of a UCP1-independent 'antiobesity factor' from BAT, but that 'such an antiobesity factor remains hypothetical'[17]. Interestingly, the underlying UCP1-independent factors must prevent the accumulation of nutrient energy more effectively than UCP1-dependent thermogenesis at room temperature.

To identify the controlling factors and molecular networks of paradoxical obesity resistance in UCP1-deficient mice, we apply an array of technologies, ranging from mouse metabolic phenotyping to global omics analyses. Strikingly, we discover that endogenous fibroblast growth factor 21 (FGF21) is the single key regulator mediating paradoxical resistance to obesity. Consolidating this finding by generating the UCP1-FGF21 double knockout (dKO) mouse, transcriptomic profiling of multiple tissues determines FGF21-dependent white adipose tissue remodeling as the site of inefficient energy turnover, with networks suggesting enhanced lipid turnover and oxidative metabolism.

## Results

**UCP1 KO mice are protected from DIO at room temperature**. To prevent confounding effects of cold stress in early development, wildtype (WT) and UCP1 KO mice were weaned and raised at thermoneutrality and transferred from 30 to 23 °C (standard housing temperature for laboratory mice) at an age of 10–12 weeks, and normal chow diet was replaced by high fat diet (HFD). In response to the treatment, body weight gain developed significantly lower in UCP1 KO compared to WT mice, resulting in significantly lower body weights after twelve weeks of HFD feeding (Fig. 1a). Thus, the 'paradoxical resistance to DIO on HFD fed UCP1 KO mice'[21,23] appears to be a robust and reproducible phenomenon among different independent laboratories.

**Protection from DIO coincides with increased FGF21**. To get mechanistic insights on the unexplained phenomenon of obesity resistance, we focused on the time-point after three weeks of HFD feeding before changes in body weight occur (Fig. 1b; indicated by the red arrow). In a second cohort of mice, no differences in body weight, body fat or liver triglycerides were detectable between WT and UCP1 KO mice after three weeks of HFD (Fig. 1c, d). Next, we focused on FGF21, as FGF21 serum levels in UCP1 KO mice are increased at temperatures below thermoneutrality[24]. While endogenous FGF21 imposes no effects on metabolic rates and body weight during chow diet feeding[24,25], its role for systemic metabolism is established under conditions of obesity, where administration of exogenous FGF21 (or FGF21 analogs) leads to improved metabolic profiles in mice and humans[26,27]. After three weeks of HFD feeding at 23 °C, we detected significantly increased serum levels of endogenous FGF21 in UCP1 KO mice (Fig. 1e), which coincide with increased FGF21 gene expression in BAT and inguinal white adipose tissue (iWAT), but not in the liver (Fig. 1f).

**FGF21 entirely controls DIO resistance in UCP1 KO mice**. Next, we generated UCP1/FGF21 double knockout (dKO) mice to evaluate the role of FGF21 signaling during HFD-induced body weight development in UCP1 KO mice. WT, FGF21 KO, UCP1 KO, and dKO mice were weaned and raised at 30 °C. At the age of 10–12 weeks, all mice were transferred to 23 °C and normal chow diet was replaced by HFD for either three (Fig. 2a, SAC1) or twelve weeks (Fig. 2a, SAC2). Impressively, dKO mice displayed identical body weight development as found for WT and FGF21 KO mice under HFD conditions, while UCP1 KO mice stayed lean as expected (Fig. 2b). The body weight gain of WT, FGF21 KO, and dKO mice was caused by the increase in fat mass, but not fat-free mass (FFM) (Fig. 2c, d), reflected in increased epididymal adipose tissue (eWAT) and iWAT mass (Fig. 2e, Supplementary Fig. 1a, b). In contrast, BAT mass of UCP1 KO and dKO mice was increased independently of body fat status and FGF21, and thus, was solely driven by the lack of UCP1 (Fig. 2e, Supplementary Fig. 1c). Lower liver triglyceride content in UCP1 KO mice was a consequence of the lean phenotype, as no differences were visible after three weeks of HFD (Fig. 2f). Glucose tolerance was not different between the genotypes (Fig. 2g) but UCP1 KO mice were more insulin-sensitive, indicated by lower basal blood glucose, insulin levels, and HOMA-IR (Fig. 2h–j). Systemic lipid metabolism was characterized by lower leptin levels in UCP1 KO mice after twelve weeks of HFD (Fig. 2k), possibly due to lower fat mass (Fig. 2c). While serum free fatty acids were not affected (Fig. 2l), increased triglyceride concentrations associate to the lack of FGF21 after twelve weeks of HFD (Fig. 2m). Importantly, the paradoxical resistance of UCP1 KO mice to DIO appeared only at 23 °C. Maintaining animals

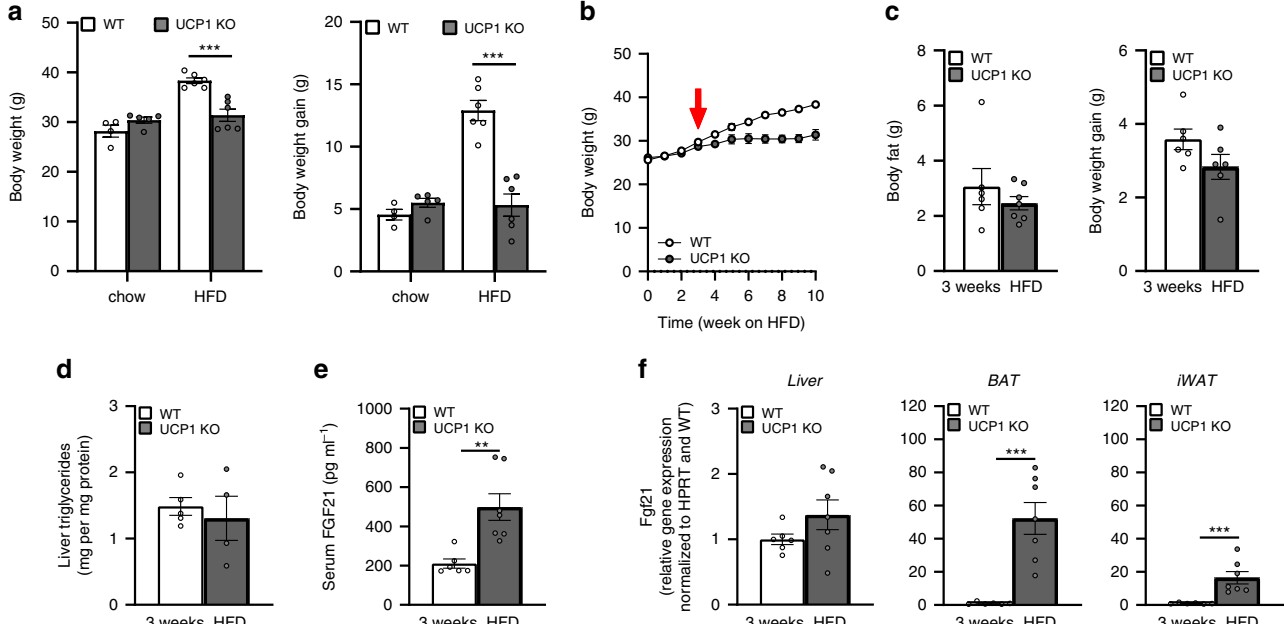

**Fig. 1 UCP1 KO mice display resistance to DIO and increased FGF21 at room temperature.** At the age of 10–12 wks, WT and UCP1 KO mice were transferred from 30 °C to room temperature and high fat diet (HFD) or chow diet was offered for 12 wks. **a** Mean body weight and body weight gain after 12 wks chow or HFD treatment. **b** Body weight trajectory at 23 °C on HFD. After 3 wks of HFD feeding at 23 °C, **c** body fat content and body weight gain, **d** liver triglycerides, **e** serum FGF21 levels, and **f** FGF21 gene expression in liver, BAT and inguinal WAT was assessed. **a**: $n = 4/5/6/6$ WT chow/UCP1 chow/WT HFD/UCP1 KO HFD; **b**: $n = 6/6$ WT/ UCP1 KO; **c**: $n = 6/7 – n = 6/6$; **d**: $n = 4/5$; **e**, **f**: $n = 6/7$). Data are presented as mean ± SEM. Data were analyzed by Student's $t$-test **$P < 0.01$, ***$P < 0.001$. Source data are provided as a Source Data file.

during the twelve weeks of HFD intervention at thermoneutral conditions (30 °C) led to identical body weight and body fat accumulation in all genotypes (Fig. 2n, o), coinciding with the lack of FGF21 induction in BAT and iWAT (Supplementary Fig. 1d, e). Collectively, these data delineate that increased levels of FGF21 entirely mediate the paradoxical resistance to DIO of UCP1 KO mice at room temperature.

**Mouse metabolic phenotyping.** Next, we performed comprehensive mouse metabolic phenotyping to identify the parameters of energy metabolism that are responsible for obesity resistance. Using indirect calorimetry, no genotypic differences in metabolic rate were found when normalized to fat free mass (Fig. 3a) or when plotted against whole body weight (Fig. 3b). The use of fat free mass eliminates the confounding impact of differences in inert fat mass (Fig. 2c, Supplementary Fig. 2a, b). Food intake and water intake were normalized to fat free mass (Fig. 3c, d) and given per animal (Supplementary Fig. 2c, d). Food intake was not significantly different between the genotypes. Activity counts (Fig. 3e) and body temperature (Fig. 3f) did not differ significantly between genotypes. In contrast to previous findings[23], no shift in respiratory exchange ratio (RER, Fig. 3g) and its percent relative cumulative frequency analysis (PRCF; Fig. 3h) were detected. Recent studies associate the protection against obesity development to the cold stress-microbiome axis,[28] affecting bile acid (BA) composition and energy assimilation[28–30]. The reduction of ambient temperature from 30 to 23 °C represents cold stress for UCP1 KO mice. Thus, we performed a kinetic experiment and analyzed the microbiota and bile acids of mice after three weeks (Fig. 3i–o) and twelve weeks of HFD feeding (Supplementary Fig. 2e–j). To investigate how diet and ambient temperature affect the microbiota phylogenetic richness in each caecum sample, we analyzed the alpha -diversity assessed by rarefaction and phylogenetic diversity, followed by principal component analysis (PCA) of unweighted UniFrac distances between

the caecum samples from the different groups. Analysis at the phylum and family level indicated that the microbiota was dominated by Firmicutes (Fig. 3k, Supplementary Fig. 2g). No differences between the genotypes were detected neither before the onset of obesity development (Fig. 3i–l) nor after twelve weeks of HFD feeding (Supplementary Fig. 2e–h), suggesting that the microbiome has no impact on the lean phenotype of UCP1 KO mice. Furthermore, differences of the bile acid profile were minor between genotypes (Fig. 3m–o, Supplementary Fig. 2i, j). Feces energy content and food assimilation, assessed by bomb calorimetry, did not differ significantly between UCP1 KO and dKO mice (Fig. 3p, q).

**Urine composition and excreted energy.** Metabolic phenotyping revealed increased water intake of UCP1 KO mice (Fig. 3d, Supplementary Fig. 2d), supporting recent reports on induced drinking by FGF21 administration or endogenous FGF21[31–33]. Thus, increased endogenous levels of FGF21 impose physiological consequences in UCP1 KO mice. The increased water intake of UCP1 KO mice (Fig. 3d) was corroborated in a second cohort of mice (Fig. 4a) that also served to collect urine. We discovered that urine excretion of UCP1 KO mice was increased (Fig. 4a). To investigate whether energy loss may occur via energy-storing metabolites in urine as suggested before[34], we measured urine energy content using bomb calorimetry showing that excessive energy is not lost via the urine (Fig. 4b). Urine creatinine and albumin concentration were not elevated in UCP1 KO mice (Fig. 4c, d), excluding kidney failure as cause for increased water intake, which is in line with previous observations[32]. While targeted quantitative metabolomics reveals no changes in hexose concentration (Fig. 4e), an increased secretion of acyl-carnitines and amino acids in the urine of UCP1 KO mice was observed, indicative of altered metabolism (Fig. 4f, g; the full list of metabolites can be found in Supplementary Table 1).

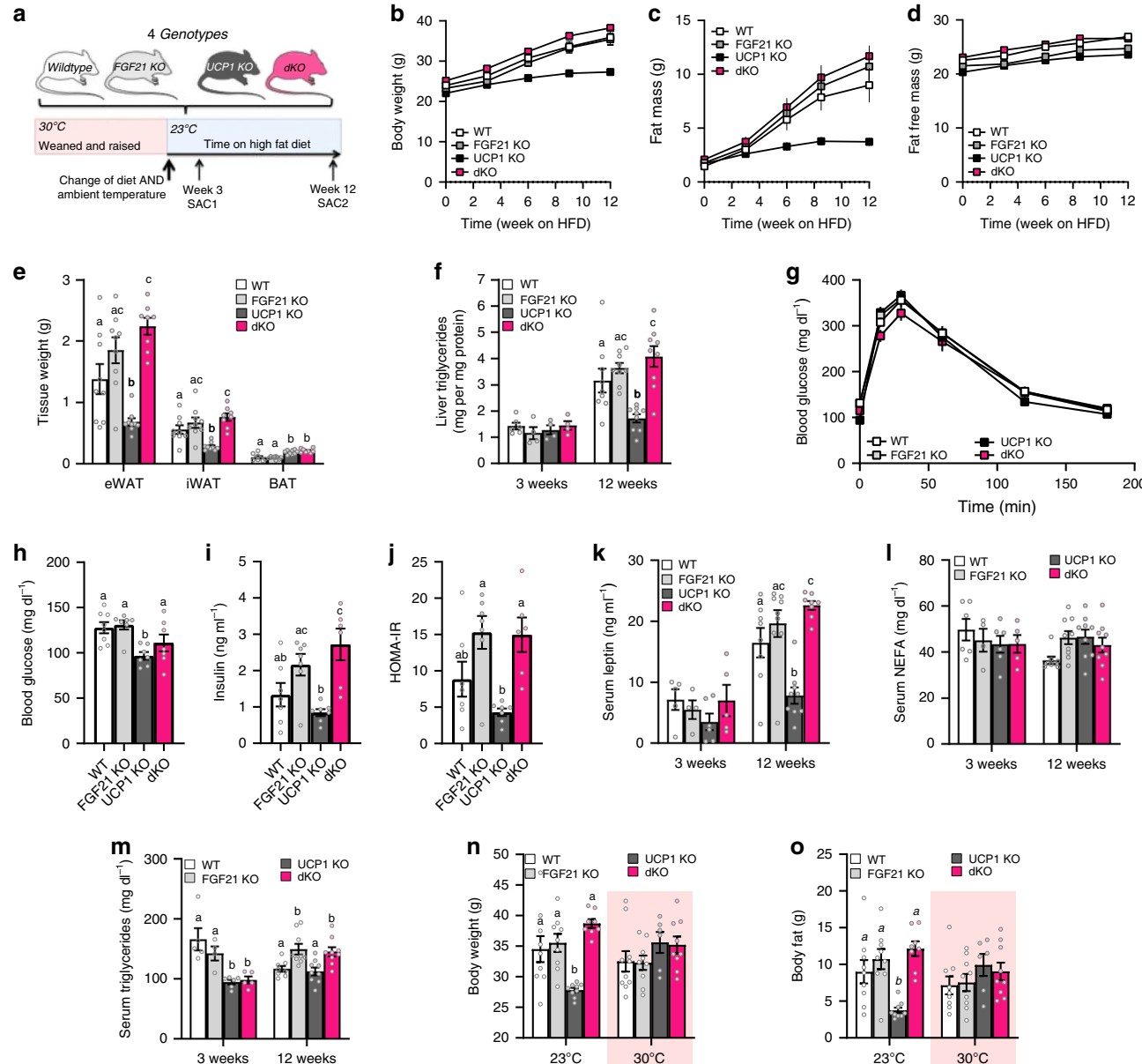

**Fig. 2 FGF21 entirely controls the resistance to diet-induced obesity in UCP1 KO mice. a** Scheme depicting the study design (SAC: sacrifice). At the age of 10–12 wks, WT, FGF21 KO, UCP1 KO and UCP1/FGF21 dKO mice were transferred from 30 to 23 °C and fed a high fat diet (HFD) for 3 or 12 wks. Trajectories of **b** body weight, **c** body fat and **d** fat free mass. **e** Weights of epididymal and inguinal WAT; and BAT. **f** Liver triglycerides after 3 and 12 wks of HFD. **g** Glucose tolerance test after 8 wks of HFD (4 h food withdrawal); **h** basal blood glucose, **i** insulin levels and **j** HOMA-IR. **k** Serum leptin, **l** NEFA and (**m**) triglyceride (TG) levels after 3 and 12 wks of HFD. **n** Mean body weight and **o** body fat of mice fed HFD for 12 weeks at 23 or 30 °C. **b**–**e**: $n = 9/9/9/9$ WT/FGF21 KO/UCP1 KO/dKO; **g**: $n = 8/8/9/9$; **n**: $n = 8/7/7/7$; **l**, **j**: $n = 7/7/7/6$; **f**: $n = 6/4/4/4 – 9/9/9/9$ 3 wks WT/FGF21 KO/UCP1 KO/dKO – 12 wks WT/FGF21 KO/UCP1 KO/dKO; **k**–**l**: $n = 5/4/6/5 – 8/9/9/9$; **n**: $n = 9/9/9/9 – 10/9/6/9$ 23 °C WT/FGF21 KO/UCP1 KO/dKO – 30 °C WT/FGF21 KO/UCP1 KO/dKO; **o**: $n = 9/9/9/8 – 9/9/6/9$. Data are presented as mean ± SEM. Data were analyzed by one-way ANOVA followed by Bonferroni test. Different letters indicate significant differences ($P < 0.05$). Source data are provided as a Source Data file.

**Transcriptional profiling reveals FGF21-controlled browning.** To gain mechanistic insights on the molecular basis for changed metabolism and how FGF21 controls obesity resistance, next-generation RNA sequencing of all genotypes was performed in metabolically active tissues (BAT, iWAT, liver, and muscle) after three weeks of HFD (Fig. 5), before the divergence of body weights (Fig. 2b). Hierarchical clustering of significantly regulated genes in BAT depicted two distinct gene clusters showing similar expression patterns of WT/ FGF21 KO versus UCP1 KO/ dKO samples (Fig. 5a). Strikingly, massive differential gene regulation in iWAT of the UCP1 KO mouse was reversed by additional ablation of FGF21 in the dKO mouse (Fig. 5b), demonstrating FGF21-controlled tissue remodeling. Given 912 differentially expressed genes (DEGs) in iWAT (Supplementary Fig. 3a), about 90% of the DEGs were upregulated only in UCP1 KO mice (Fig. 5b). The number of DEGs in liver and muscle were relatively minor (Fig. 5c, d, Supplementary Fig. 3a). The Venn analysis of DEGs between UCP1 and dKO mice depicts that FGF21-mediated gene expression almost exclusively occurs in iWAT (Fig. 5e). To investigate the involvement of secreted factors other than FGF21, we looked at the gene expression of commonly known batokines[35,36]. Some of these appear to be regulated by the

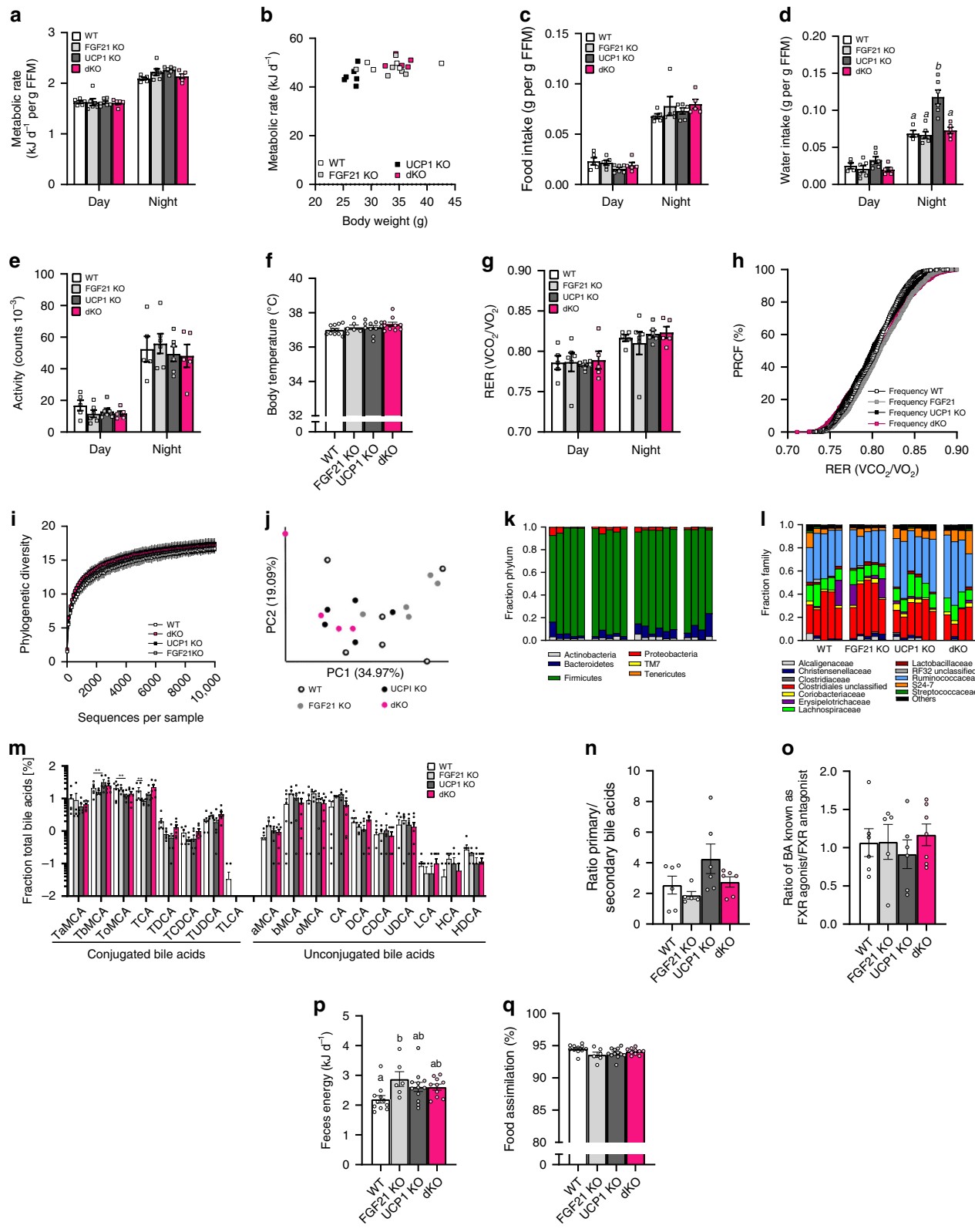

loss of UCP1, but none associates to obesity resistance, as their regulation was not differentially affected in dKO mice (Fig. 5f). Well-established and recently identified 'browning' markers[37,38] were exclusively upregulated in iWAT of UCP1 KO mice (Fig. 5g), and 'browning' reprogramming, even in the

absence of UCP1, is further supported by comparing the transcriptional profile to a previously published 'browning signature'[37] (Supplementary Fig. 3b, c). Based on the regulatory network of the molecular browning signature[37], the control over browning in UCP1 KO mice appears to be predominantly

**Fig. 3 Phenotyping of mouse energy metabolism.** After 9 wks of HFD feeding at room temperature, indirect calorimetry was performed with WT, FGF21 KO, UCP1 KO, and UCP1/FGF21 dKO mice. **a** Energy expenditure per fat free mass (FFM) and **b** correlation of body weight vs energy expenditure per animal, **c** day/night food intake per fat free mass, **d** day/night water intake per fat free mass, **e** day/night activity, **f** body temperature and **g** day/night respiratory exchange ratio (RER) with **h** percent relative cumulative frequency analysis (PRCF). After 3 wks of HFD feeding the gut microbiota and microbial metabolism were assessed. **i** Alpha rarefaction plot displaying species richness dependent on sampling depth (max depth = 9410 sequences per sample). **j** Principal coordinate analysis (PCA, weighted UniFrac) displaying β-diversity of the gut microbial community. The percentage of the variation explained by the plotted principal coordinates is indicated in the axis labels. Each dot represents a caecal community. Relative abundance at **k** phylum and **l** family level in caecal community of mice, displaying only relative abundances >1%. **m** Plasma bile acid (BA) profile and **n** ratio of primary to secondary BAs, and **o** ratio of BAs known as FXR agonist/FXR antagonist. **p** Feces energy content and **q** percent food assimilation were measured after 6 weeks of HFD feeding. **a**, **b**, **e**, **g**, **h**: n = 5/6/6/5 WT/FGF21 KO/UCP1 KO/dKO; **c**: n = 5/5/6/5, **d**: n = 4/6/6/5; **i–l**: n = 5/5/6/4; **m**, **n**: n = 6/5/6/7; **o**: n = 6/5/6/5; **f**, **p**, **q**: n = 11/6/12/10. Data (**a–g**, **p**, **q**) were analyzed by one-way ANOVA, followed by Bonferroni test and are presented as mean + SEM. Different letters indicate significant differences (P < 0.05). Data are presented as mean ± SD (**i**) or mean ± SEM (**m–o**) with significant differences shown as **P < 0.01. Source data of **a–h**, **m–q**, **h** are provided as a Source Data file.

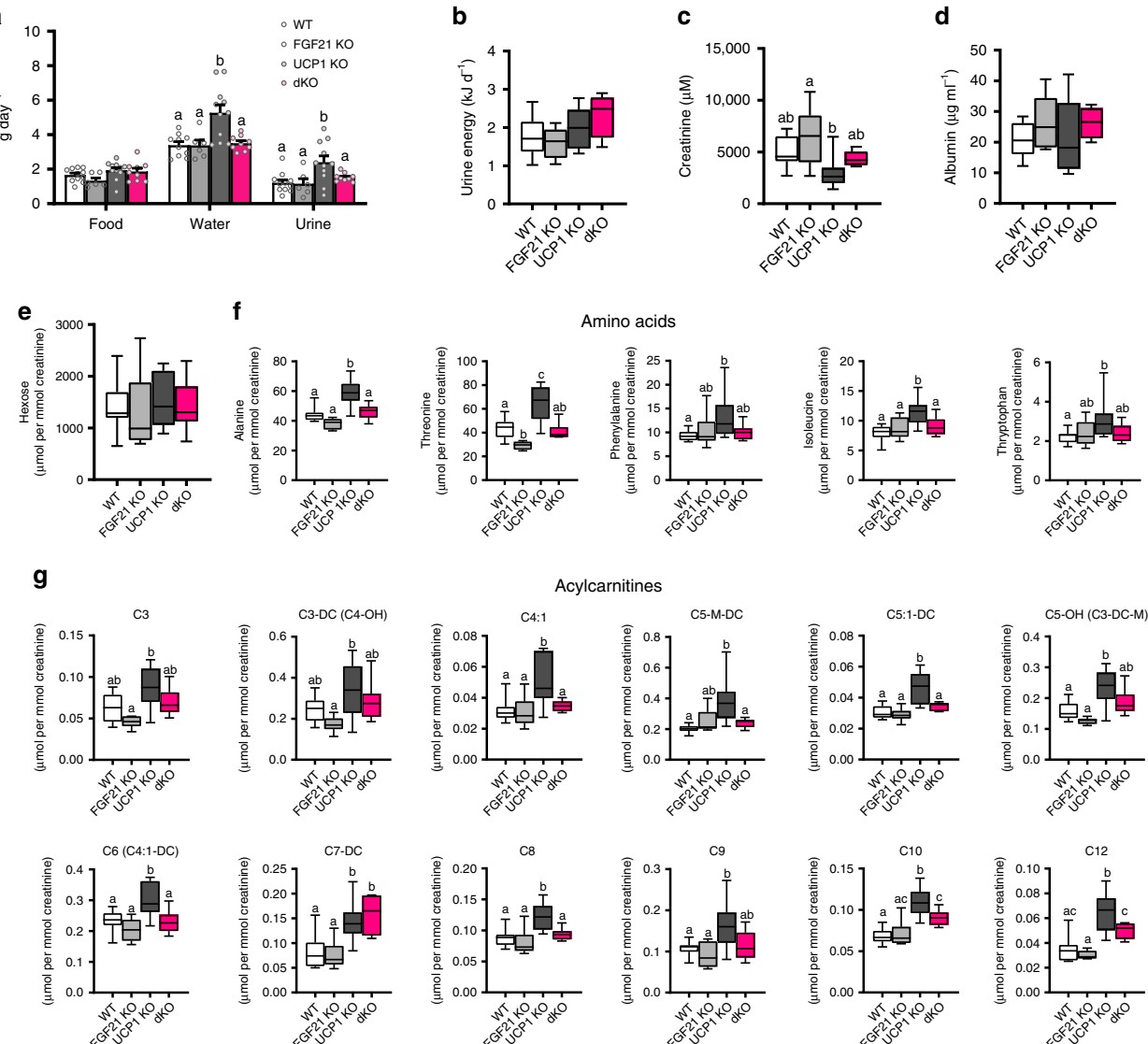

**Fig. 4 Urine composition and excreted energy.** After 6 wks of HFD feeding at room temperature, **a** food and water intake were determined and urine was collected. **b** Urine energy content, **c** urine creatine concentration, **d** urine albumin concentration, **e** urine hexoses, **f** urine amino acids, and **g** urine acylcarnitines were measured. Data are presented as box-plots, showing the metabolite distribution. The central mark indicates the median with the bottom and top edges of the box indicating the 25th and 75th percentiles, respectively. The whiskers extend to the most extreme data points. **a** (food and Urine), **b**, **c**, **f** (except Threonine), **g**: n = 11/6/11/9 WT/FGF21 KO/UCP1 KO/dKO; **a** (water): n = 10/6/11/9; **d**: n = 6/6/9/6; **e**: n = 11/6/10/9; **f**: n = 11/5/10/9; **f** (Threonine): n = 11/5/10/9. Data were analyzed by one-way ANOVA followed by Bonferroni test with different letters indicating significant differences (P < 0.05). Source data are provided as a Source Data file.

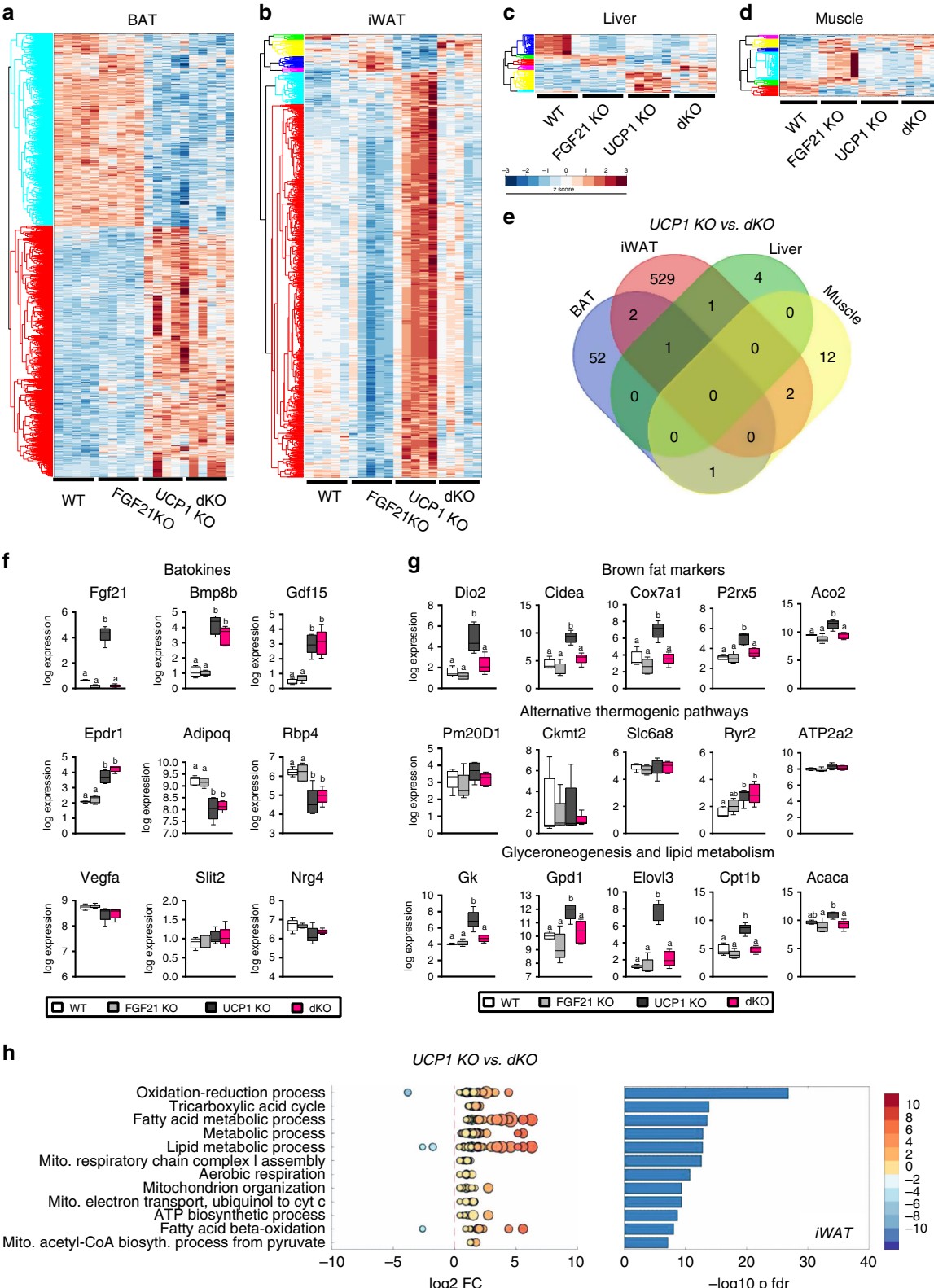

regulated by PPAR alpha and PGC1 alpha (Supplementary Fig. 3d). At least on the transcriptional level, genes of recently suggested UCP1-independent thermogenic pathways, such as endogenous uncoupler synthesis[39], creatine metabolism[40] and calcium cycling[41] were not differentially expressed, while genes of glyceroneogenesis and lipid metabolism were highly induced in UCP1 KO mice (Fig. 5g).

The gene ontology (GO) enrichment analysis of the DEGs in iWAT uncovers an unbiased, detailed picture of molecular network enrichments in UCP1 KO mice compared to dKO mice,

**Fig. 5 Transcriptional profiling reveals FGF21-controlled browning of inguinal fat.** After 3 wks of HFD feeding at room temperature, organs were harvested for transcriptomic analysis. Heat maps illustrate all significantly expressed transcripts (adjusted *p*-value < 0.01) in **a** BAT, **b** iWAT, **c** liver and **d** muscle. Colors refer to gene expression z-score (**b–d**). **e** Venn diagram of all DEGs between UCP1 KO and dKO mice (adjusted *p*-value < 0.01). **f**, **g** Box plots showing the distribution of log2 gene expression level. The central mark indicates the median, and the bottom and top edges of the box indicate the 25th and 75th percentiles, respectively. The whiskers extend to the most extreme data points. **f** Batokine expression in BAT and in WAT, **g** marker genes of 'browning' (upper panel), genes associated with UCP1-independent thermogenesis (middle panel) and genes involved in glyceroneogenesis and lipid cycling (lower panel). **h** GO enrichment analysis of significantly regulated genes (adjusted *p*-value < 0.01) in iWAT of UCP1 KO vs. dKO. The top 12 regulated pathways are shown. Dot plots display expression of single genes related to the enriched pathways. Dot color refers to log2 FoldChange, while dot size refers to −log10 significance of regulation. Bar plots display significance of pathway enrichment. *n* = 5/5/5/5 WT/FGF21 KO/UCP1 KO/ dKO. Data (**f**, **g**) were analyzed by one-way ANOVA followed by Bonferroni test with different letters indicating significant difference (*P* < 0.05).

showing association of upregulated pathways to mitochondrial oxidation and lipid metabolism (Fig. 5h). The increased mobilization of lipids is further supported by morphological changes in iWAT, which shows increased multilocularity in UCP1 KO mice (Supplementary Fig. 4a). Notably, the browning of iWAT was specific for UCP1 KO mice, associating browning to the lean phenotype. In contrast to iWAT, BAT transcriptomics revealed enriched pathways very similar in UCP1 KO and dKO mice (Supplementary Fig. 4b), demonstrating that BAT reprogramming is predominantly driven by the lack of UCP1, with only minor impact of FGF21. Almost no differences in metabolic pathways between all genotypes were seen in liver and muscle (Supplementary Fig. 4b). Taken together, the global transcriptomic analyses exclude BAT as contributing organ to the development of obesity resistance in UCP1 KO mice and highlights iWAT remodeling as driving force. The molecular classification supported by clustering and enrichment analyses suggests enhanced metabolic activity of iWAT in UCP1 KO mice, that has been also annotated as the 'browning of white fat'.

## Discussion

The knockout mouse of thermogenic UCP1 provides probably the most extensively studied animal model to investigate anti-obesity mechanisms mediated by BAT. However, these mice express the phenomenon of paradoxical resistance to DIO at room temperature, clouding the interpretation on brown fat significance in obesity on one hand, but also suggesting the existence of alternative pathways of energy loss on the other. Using UCP1/FGF21 dKO mice, we show that FGF21 mediates the resistance to DIO and the remodeling of iWAT ('browning') in UCP1 KO mice. Therefore, this study solves the molecular identity of primary endocrine control during UCP1-independent (previously paradoxical) obesity resistance, opening a window to identify novel approaches for obesity therapy and to distinguish UCP1-dependent and -independent contributions to obesity resistance.

Given the importance of therapeutically treating the obesity epidemic, the underlying mechanisms of energy loss in the presence and absence of UCP1 are indisputably important. While a substantial body of evidence emphasizes the importance of increasing BAT mass and UCP1 expression to counteract obesity[3,4], it should be noted that the UCP1 KO mouse model only supports anti-obesogenic effects of BAT in thermoneutral conditions[9]. The pioneering metabolic studies on paradoxical resistance to obesity in UCP1 KO mice were in line with our metabolic assessment showing no significant differences in energy metabolism after dietary change[21] and after 2 and 17 weeks of high fat diet feeding[23]. In our study, however, we completed the bioenergetic assessment by including feces and urine energy content. Our comprehensive phenotyping enabled us to search for differences in energy balance based on the assumption that the murine system obeys the first law of thermodynamics and no energy disappears. Given the absolute values of energy intake

(daily food intake) and output (daily energy expenditure, feces energy, and urine energy) (Supplementary Table 2, raw data), we calculated the net energy balance of each genotype (Supplementary Table 2, calculated energy balance). As compared to UCP1 KO mice, the other genotypes display positive energy balance based on the combination of all bioenergetic parameters. The surplus of ~1–2 kcal (1 kcal = 4.2 kJ) per day over the 8 weeks period would lead to fat accumulation that minorly differs from the experimentally assessed fat mass values (Fig. 2c). With the technically challenging measurements of urine energy content, we assume that almost all nutrient energy is detected, enabling us to partition 100% nutrient energy into the energy containing/dissipating modules (Supplementary Table 2). Here, it transpires that WT, FGF21 KO, and dKO mice ingest more energy than they lose, whereas UCP1 KO mice fully dissipate energy intake. Importantly, our bioenergetic calculations emphasize that minor imbalances in daily energy metabolism can accumulate to substantial differences in obesity development. This requires consideration in studies addressing human obesity, which cannot simply be attributed to single changing parameters. In line with our findings, other studies with comprehensive mouse phenotyping supports this notion showing that single parameters such as energy intake or expenditure provide only minor differences leading to exessive fat storage[42].

The UCP1/FGF21 dKO mouse unambiguously pinpoints endogenous FGF21 as primary crucial mediator of obesity resistance, and iWAT as the target organ. These observations are coherent with the pharmacology of exogenous FGF21 that does not require UCP1-dependent thermogenesis for beneficial metabolic effects during obesity[43,44]. While mild cold is required as the trigger of FGF21 release, this hormone fully unfolds its anti-obesity effects under high-fat diet conditions. In previous studies, others showed that BAT becomes a source of FGF21 in the cold[45], which is further potentiated in UCP1 KO mice[24]. The requirement to maintain body temperature under long-term cold conditions of 5 °C overwrites FGF21 action, seen as FGF21-independent browning in UCP1 KO mice[25].

While previous morphological observations in iWAT of UCP1 KO mice suggested browning of iWAT[23], this study provides with the transcriptomic analysis an important resource for researchers investigating UCP1-independent molecular networks of obesity relevance and the molecular landscape for FGF21-dependent browning. FGF21-dependent intracellular signaling is mediated by FGFRs in combination with beta klotho[46], but other unknown factors may contribute. While we did not investigate unknown factors of intracellular FGF21 signaling, the genetic network underlying browning and metabolic consequences in iWAT were addressed in silico to get further insights into potential pathways. The recently published molecular browning signature and its regulation, that is based on meta-analysis of more than 100 published data sets[37], has been adopted to our data for prediction of regulatory pathways. This analysis reveals the molecular induction of browning, controlled by PPAR alpha and PGC1

alpha, which were more pronouncedly increased than NR4A1, an adrenergically responsive gene (see Supplementary Fig. 3d). Mapping these transcription factors on highly significant DEGs of iWAT in UCP1 KO mice elucidates routes for the induction of lipid metabolism genes, such as Cpt1b and Pdk4. This is further supported by an unbiased pathway analysis of all transcriptomic data, revealing the FGF21-dependent enrichment of lipid and oxidative metabolism only in iWAT, including the typical browning genes[37]. These observations are coherent with pharmacological studies administering FGF21[47]. Thus, enhanced lipid metabolism may prevent obesity, as suggested for other mouse[47] and clinical studies[48].

FGF21-dependent browning in DIO resistant UCP1 KO mice also raises the question on alternative thermogenic mechanisms. The simultaneous induction of lipolysis and lipogenesis fuels the idea of lipid futile cycling as an ATP-dependent thermogenic mechanism[49–51], which may serve to maintain the lean phenotype of the UCP1 KO mouse. Other recently suggested mechanisms, such as creatine and calcium futile cycling[40,41], were induced in cold-acclimated UCP1 KO mice but the pronounced induction of gene expression was not seen in this study. This is probably not surprising as conditions of mild cold with overexcessive energy intake impose less thermoregulatory constraints.

Not obviously affecting energy expenditure, food intake or body temperature, the thermogenic potential of FGF21-controlled browning of iWAT may be weak as compared to BAT, but the metabolic impact appears to be powerful for preventing obesity. Seminal findings by others demonstrated FGF21's potency to directly brown WAT[52,53], but whether FGF21 impacts browning also indirectly in the UCP1 KO mouse, remains to be determined.

Our data reveal no significant impact of endogenous FGF21 on food consumption. In contrast, high doses of exogenous FGF21 reduce food intake of obese UCP1 KO mice[43]. However, pair-feeding experiments also reveal that this only partially explains the body weight reduction[43]. In WT mice, increased levels of FGF21 are usually associated with increased energy expenditure but no differences in food intake, as seen during FGF21 gene therapy[54], by transgenic overexpression of FGF21[55] and exogenous FGF21 administration[26]. Recently published effects on drinking behavior were evident in our metabolic phenotyping, demonstrating physiological significance of endogenous FGF21. The subsequent higher urine excretion, however, did not reveal significant impact on energy loss by increased secretion of amino acid and lipids, but hinted towards altered metabolism in the UCP1 KO mouse.

Beyond the UCP1 KO model, the control of obesity resistance by adipose FGF21 may also explain the lack of obesity resistance in other important BAT-deficient mouse models, such as mice lacking ß-adrenergic receptors (ß-less) or mice with isolated BAT deficiency (UCP1-DTA)[6,7]. In the ß-less mice, the adrenergic stimulation of adipose tissue that stimulates FGF21 release, should be missing[7], while in BAT-less mice, there is no FGF21-release from BAT[24]. Collectively, endogenous FGF21-signaling is an interesting therapeutic target, as it appears sufficient to prevent obesity in the absence of UCP1.

## Methods

**Animals**. The experiments were performed in homozygous male WT and UCP1 KO mice (genetic background C57BL/6J). We reduced confounding developmental adaptation to thermal stress by breeding, raising and maintaining all mice at thermoneutrality before the experimental procedures. Mice were housed in groups with ad libitum access to food and water, and a 12:12-h dark–light cycle (lights on:7:00 CET). At the age of ~10–12 weeks, mice were randomly assigned to chow (Altromin) or 58% high fat diet (Research diets, D12331) fed groups, and housing temperature was changed from 30 °C to 23 °C+/−1 °C. After 3 weeks and after 12 weeks of dietary intervention, mice were sacrificed 3–4 h after lights went on, and serum and tissue samples were collected. Identical experiments were

performed with male WT, FGF21 KO, UCP1 KO and UCP1/FGF12 double KO (dKO) mice (genetic background C57BL/6J). The breeding strategy to generate the double KO mice was as follows: The first breeding step generated heterozygous UCP1/FGF21 KO mice by breeding homozygous UCP1 KO with homozygous FGF21 KO mice. Afterwards, those mice were mated to generate homozygous mice for all genotypes, which were again mated and used for the experiment. The animal experiments complied with all ethical regulations for animal testing and research, including animal maintenance and experimental procedures, that the animal welfare authorities of the local animal ethics committee of the state of Bavaria (Regierung Oberbayern) approved in accordance with European guidelines.

**Gene expression analysis**. RNA was extracted using Qiazol according to the manufacturer's instructions (Qiazol Lysis Reagent, Qiagen, Germany). Synthesis of cDNA and DNase treatment were performed from 1 μg of total RNA using QuantiTect Reverse Transcription Kit (Qiagen, Germany). Quantitative real-time PCR was performed on the ViiA™ 7 Real-Time PCR System (Applied Biosystems, USA). The PCR mix contained SybrGreen Master Mix (Applied Biosystems, USA), cDNA corresponding to 5 ng of RNA, and gene specific primer pairs. Gene expression was calculated as ddCT using HPRT for normalization. The data are shown as values relative to the WT group. The oligonucleotide primer sequences are available in Supplementary Table 3.

**Indirect calorimetry and metabolic cages**. Energy expenditure, food intake, activity and respiratory exchange ratio ($RER = VCO_2$ produced/$VO_2$ consumed) were measured by indirect calorimetry using an open respirometry system with simultaneous measurements of activity, food and water intake (TSE PhenoMaster, TSE Systems, Germany). Measurements were performed in 10 min intervals over a period of 4 days. The last 2 days were analyzed to assess genotypic differences. Mice were house individually in metabolic cages allowing the monitoring of food and water uptake, as well as feces and urine collection (Tecniplast, Hohenpeissenberg, Germany). After 2 days of acclimation, 24-h food and water intake were determined by daily weighing of food hoppers and water bottles. Feces and urine were collected in the same time intervals. Body mass was monitored by daily weighing of mice. Urine samples of three 24-h cycles were transferred to Eppendorf tubes, centrifuged at $5000 \times g$ for 10 min. Aliquots for subsequent analysis were stored at −20 °C.

**Bomb calorimetry**. Feces samples (about 1 g each) were dried at 60 °C for two days, homogenized in a grinder and condensed with pressure to determine energy content by bomb calorimetry (IKA C7000, Staufen, Germany). To obtain percent of food assimilated energy, the energy content of feces was subtracted from energy uptake which was set to 100%. To determine the amount of energy lost via urine, 1 ml of thawed urine was loaded on cotton coils as combustion aid (Cotton Cell Rolls 8 mm, Henry Schein Inc., Melville, NY, USA) that had been desiccated to constant weight at 60 °C, prior to loading[56]. The exact volume could be determined by using high precision pipettes. The energy content of unloaded cotton coils was 16.918 kJ g$^{-1}$ dry weight. Loaded coils were then carefully handled and desiccated again to constant weight at 60 °C in a drying oven. The weight difference between the unloaded dry cotton coil and the loaded dry cotton coil accounted for the dry residues contained in the respective volume of loaded urine. The loaded cotton coils were then combusted in a bomb calorimeter (IKA C7000, Staufen, Germany). The energy content of the dry urine residue was automatically calculated by the software of the bomb calorimeter, taking into account the total supplemented energy from both the cotton coil and the fuse, which was needed to initiate the combustion of the sample. We then normalized urine residue energy content to 1 ml of urine. To calculate daily energy loss, normalized urine energy content was multiplied by the total amount of excreted urine.

**Serum analysis**. For the analysis of serum FGF21, Leptin, NEFA, Triglycerides, as well as Insulin, commercially available assay kits were used according to the manufacturer's recommendations (Intact FGF-21 ELISA Kit, Eagle Biosciences; Mouse Leptin ELISA Kit, Crystal Chem; NEFA-HR2 Wako, LabAssay Triglyceride Wako and Insulin Elisa, Alpco).

**Liver triglycerides**. The triglyceride concentration of liver samples was measured after extraction with 10 mmol l$^{-1}$ sodium phosphate buffer (pH 7.4) containing 1 mmol l$^{-1}$ EDTA and 1% polyoxyethylene (10) tridecylether using a Triglyceride Kit according to the manufacturer's recommendations (LabAssayTM Triglyceride, Wako).

**Glucose tolerance test**. After 8 weeks of dietary intervention, an intra-peritoneal (i.p.) glucose tolerance test (GTT) was performed. Glucose (2 mg g$^{-1}$ of body weight) was applied i.p. 4 h after food withdrawal. Glucose levels were measured before, 15, 30, 60, and 120 min after glucose application.

**Transcriptomic analysis**. Total RNA was extracted from inguinal WAT, BAT, liver and muscle of WT, FGF21 KO, UCP1 KO and dKO mice ($n = 5$) using Qiazol according to the manufacturer's instructions (Qiazol Lysis Reagent, QIAGEN). The quality of the RNA was determined with the Agilent 2100 BioAnalyzer (RNA 6000

Nano Kit, Agilent). All samples had an RNA integrity number (RIN) value greater than 8. For library preparation, 1 μg of total RNA per sample was used. RNA molecules were poly(A) selected, fragmented, and reverse transcribed with the Elute, Prime, Fragment Mix (EPF, Illumina). End repair, A-tailing, adaptor ligation, and library enrichment were performed as described in the TruSeq Stranded mRNA Sample Preparation Guide (Illumina). RNA libraries were assessed for quality and quantity with the Agilent 2100 BioAnalyzer and the Quant-iT Pico-Green dsDNA Assay Kit (Life Technologies). Strand-specific RNA libraries were sequenced as 100 bp paired-end runs on an Illumina HiSeq4000 platform. The STAR aligner* (v 2.4.2a)[57] with modified parameter settings (–twopassMode = Basic) was used for split-read alignment against the mouse genome assembly mm10 (GRCm38) and UCSC known Gene annotation. To quantify the number of reads mapping to annotated genes we used HTseq-count° (v0.6.0). Raw read counts were normalized and DEG were estimated using R package DESeq2. Genes with a maximum expression of 15 counts or below were removed from the data. We further removed genes with a cumulative count sum of 75 or below as well as genes that showed no expression at all in more than 10 samples. Hierarchical clustering was performed using 'Euclidean' distance measure and nearest distance linkage method. Significance of Gene Ontology enrichments were estimated using a hypergeometrical distribution test. All calculations were done using R version 3.4 and Matlab R2018a.

All RNA Seq data have been deposited into the gene expression omnibus (GEO) under the accession number GSE122167.

### DNA extraction and 16S rRNA gene sequencing.
Mouse fecal pellets were collected and preserved at −80 °C until processing. DNA extraction and 16S rRNA sequence analysis, genomic DNA from fecal pellets was extracted by repeated bead-beating using a Fast-Prep System with Lysing Matrix E (MPBio, CA). followed by ammonium acetate precipitation, and purification by QIAmp DNA mini kit (Qiagen, Germany). Bacterial DNA was profiled by sequencing of the V4 region of the 16S rRNA gene on an Illumina MiSeq (llumina RTA v1.17.28; MCS v2.5) using 515F and 806R primers designed for dual indexing[58] and the V2 kit (2 × 250 bp paired-end reads). Samples were amplified in duplicates in reaction volumes of 25 μl containing 1x Five Prime Hot Master Mix (Quantabio, MA), 200 nM of each primer, 0.4 mg ml$^{-1}$ BSA, 5% DMSO and 20 ng of genomic DNA. PCR was carried out under the following conditions: initial denaturation for 3 min at 94 °C, followed by 25 cycles of denaturation for 45 s at 94 °C, annealing for 60 s at 52 °C and elongation for 90 s at 72 °C, and a final elongation step for 10 min at 72 °C. Replicates were combined, purified with the NucleoSpin Gel and PCR Clean-up kit (Macherey-Nagel, Germany) and quantified using the Quant-iT PicoGreen dsDNA kit (Thermo Fisher Scientific). Equal amounts of purified PCR products were pooled and the pooled PCR products were purified again using Ampure magnetic purification beads (Agencourt, Danvers, MA) to remove short amplification products. Illumina paired-end reads were merged using PEAR[59], and quality filtered to remove reads that had at least one base with a q-score lower than 20 and that were shorter than 220 nucleotides or longer than 350 nucleotides. Quality filtered reads were analyzed with the software package QIIME[60] (version 1.9.1). Sequences were clustered into operational taxonomic units (OTUs) at a 97% identity threshold using an open-reference OTU picking approach with UCLUST[61] against the Greengenes reference database[62] (13_8 release). All sequences that failed to cluster when tested against the Greengenes database were used as input for picking OTUs de novo. Representative sequences for the OTUs were Greengenes reference sequences or cluster seeds, and were taxonomically assigned using the Greengenes taxonomy and the Ribosomal Database Project Classifier[63]. Representative OTUs were aligned using PyNAST[60] and used to build a phylogenetic tree with FastTree[64], which was used to calculate α- and β-diversity of samples using Phylogenetic Diversity. Chimeric sequences were identified with ChimeraSlayer[65] and excluded from all downstream analyses. Similarly, sequences that could not be aligned with PyNAST, singletons, sequences present in the blank extraction control and very low abundant sequences (relative abundance < 0.005%) were also excluded.

To correct for differences in sequencing depth, the same amount of sequences was randomly sub-sampled for each group of samples (rarefaction; maximum depth depending on sample group). A bootstrap version of Mann-Whitney-U test was used to compare the genotype-dependent abundance of OTUs at different taxonomical levels; significant differences were identified after correction for false discovery rate. Abundances higher than 1% are displayed on the genus level. QIIME was used to compute alpha diversity from rarefied OTU tables and to determine statistical significance at maximum rarefaction level by using a two-sample t-test and 999 Monte-Carlo permutations. Beta-diversity and weighted unifrac distance matrix were computed with QIIME and statistical significance of sample groupings was determined by adonis method and 999 permutations.

Microbiota 16S rDNA gene sequencing results have been deposited in the ENA sequence read archive with accession number PRJEB28632 (http://www.ebi.ac.uk/ena/data/view).

### Serum bile acids analysis.
Bile acids were analyzed using ultra-performance liquid chromatography-tandem mass spectrometry (UPLCMS/MS). Briefly, samples (50 μl) were extracted with 10 volumes of methanol, containing deuterated internal standards (d$_4$-TCA, d$_4$-GCA, d$_4$-GCDCA, d$_4$-GUDCA, d$_4$-GLCA, d$_4$-UDCA, d$_4$-CDCA, d$_4$-LCA; 50 nM of each). After 10 min of vortex and 10 min of centrifugation

at 20,000 × g, the supernatant was evaporated under a stream of nitrogen, and reconstituted in 200 μl methanol:water [1:1]. The samples were injected (5 μl) and bile acids were separated on a C18 column (1.7 μ, 2.1 × 100 mm; Kinetex, Phenomenex, USA), using water with 7.5 mM ammonium acetate and 0.019% formic acid (mobile phase A) and acetonitrile with 0.1% formic acid (mobile phase B). The chromatographic separation started with 1 min isocratic separation at 20%B. The B-phase was then increased to 35% during 4 min. During the next 10 min the B-phase was increased to 100%. The B-phase was held at 100% for 3.5 min before returning to 20%. The total runtime was 20 min. Bile acids were detected using multiple reaction monitoring (MRM) in negative mode on a QTRAP 5500 mass spectrometer (Sciex, Concord, Canada) and quantified using external standard curves.

### Targeted quantitative urine metabolomics.
The urine metabolites were analyzed with a targeted quantitative and quality controlled metabolomics approach using the AbsoluteIDQ p180 Kit (Biocrates Life Science AG). The sample preparation and analysis were performed from Biocrates (Biocrates Life Science AG). This validated assay allows the comprehensive identification and the quantification of 186 endogenous metabolites, including 21 amino acids, 19 biogenic amine, 40 ACs, 76 phosphatidylcholines (PCs), 14 lysophosphati- dylcholines (lysoPCs), 15 sphingomyelins, and the sum of hexoses (see complete list of measured amino acids, biogenic amines and ACs in Supplementary Table 2). Acylcarnitines, (Lyso-) phosphatidylcholines, sphingomyelins, and hexoses were quantified by FIA- MS/MS using a SCIEX 4000 QTRAP® (SCIEX, Darmstadt, Germany) instrument with electrospray ionization (ESI). Quantification of amino acids and biogenic amines was based on TEA (triethylamine) derivatization in the presence of internal standards followed by LC-MS/MS using a Waters XEVOTM TQ- Smicro (Waters, Vienna, Austria). The experimental metabolomics measurement technique is described in detail by patents EP 1 897 014 B1 and EP 1 875 401 B1 (S.L. Ramsay).

The reported lipid annotation represents a sum signal of all isobaric lipids with the same molecular weight (±0.5 Da range) within the same lipid class. Analyzed glycerophospholipids are differentiated according to the presence of ester and ether bonds in the glycerol moiety. The "aa" indicates that fatty acids are, at the sn-1 and sn-2 position, bound to the glycerol backbone via ester bonds, whereas "ae" indicates that fatty acids in the sn-1 or at sn-2 position are bound via an ether bond. The total number of carbon atoms and double bonds present in the lipid fatty acid chains are annotated as "C x:y", where x indicates the number of carbons and y the number of double bonds. For sphingomyelins, only fatty acids bound to the glycerol backbone at the sn-2 position are indicated under the assumption that sphingosine (d18:1) is bound at the sn-2 position. For the FIA-MS/MS analysis, MRMs are employed using lipid species-specific molecular ion and lipid class specific fragment ion. Due to the relatively low mass resolution of the triple quadrupole MS instrument, the detected MRM signal is a sum of several isobaric lipids within the same class. For example, according to LIPID MAPS data base (http://www.lipidmaps.org/), the signal of PC aa C36:6 can arise from at least 15 different lipid species that have different fatty acid compositions (e.g. PC 16:1/20:5 and PC 18:4/18:2), various positioning of fatty acid's sn-1 and sn-2 position (e.g. PC 18:4/18:2 and PC 18:2/18:4), and different double bond positions in those fatty acid chains (e.g. PC(18:4(6Z,9Z,12Z,15Z)/18:2(9Z,12Z)) and PC(18:4 (9E,11E,13E,15E)/18:2(9Z,12Z)).

### Statistical analysis.
Statistical analyses were performed using Stat Graph Prism 6 (GraphPad Software, San Diego, CA USA). Student's t-test (unpaired, 2-tailed) or one-way ANOVA and Bonferroni's multiple comparisons test were used to determine differences between the genotypes. Statistical significance was assumed at $p < 0.05$. Statistical significance of WT to other genotypes are denoted by *$p < 0.05$, **$p < 0.01$, ***$p < 0.001$. Statistical differences between the four different genotypes are indicated by letters.

### Reporting summary.
Further information on research design is available in the Nature Research Reporting Summary linked to this article.

## Data availability
All RNA sequencing data that support the findings of this study have been deposited in the National Center for Biotechnology Information Gene Expression Omnibus (GEO) and are accessible through the GEO Series accession number GSE122167. Microbiota 16S rDNA gene sequencing results have been deposited in the ENA sequence read archive with accession number PRJEB28632. The source data underlying Figs. 1a–f, 2b–o, 3a–h, 3m–q, h, 4a–g and Supplementary Figs. 1, 2 are provided as a Source Data file. All other relevant data are available from the corresponding author on request.

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

## Acknowledgements

We would like to thank Ann-Elisabeth Schwarz and Sandy Lösecke (Helmholtz Center) for excellent technical assistance, and Claudia Stöger for the project management of the GMC analysis. Furthermore, we would like to thank Valentina Tremaroli and Rozita Akrami (University of Gothenburg) for bioinformatics assistance processing the microbiota sequencing data. This work was supported partly by funding from the German Diabetes Center (DZD) to S.K., and the Swedish Research Council (2018-02150 to S.K. and 2018-03472 to M.J.). M.H.T. is supported by the Helmholtz Portofolio Program "Metabolic Dysfunction", the Alexander von Humboldt Foundation and the Helmholtz Alliance ICEMED-Imaging and Curing Environmental Metabolic Diseases. M.H.d.A. is supported by the German Federal Ministry of Education and Research (Infrafrontier grant 01KX1012), and the German Diabetes Center (DZD). The computations for microbiota composition analyses were performed on resources provided by the Swedish National Infrastructure for Computing (SNIC) through Uppsala Multidisciplinary Center for Advanced Computational Science (UPPMAX). B.S. is supported by a Marie Curie Intra European Fellowship and a Human Frontier Science Program Long-Term Fellowship. Open access funding provided by Stockholm University.

## Author contributions

S.K. and M.J. conceptualized the project, designed and performed experiments, interpreted data, and wrote the paper. D.Lu. performed bioinformatical analyses of RNASeq data, and interpreted the data with S.K. B.S. designed, performed and interpreted the microbiome analysis and M.S. performed bile acid serum profile. D.La. performed animal experiments and molecular analysis. T.S. and E.G. performed RNA sequencing. J.R. performed measurements with metabolic cages for urine/feces analyses, and interpreted the feces energy content. J.R. and H.F. designed the GMC study. M.H.T. and M.H.d.A. contributed reagents/materials/analytic tools. All authors critically reviewed and edited the paper and approved the final version of the paper.

## Competing interests

M.H.T. serves as SAB member for Novo Nordisk and ERX Pharmaceuticals Inc. All other authors declare no competing interests.
