## [Peer Review File · Nature Communications]

Reviewers' comments:

Reviewer #1 (Remarks to the Author):

1. The core observation in the manuscript by Keipert et al. is the reversion of protection against weight gain in HFD-fed UCP1-KO versus wild-type mice (at environment conditions of mild cold such as 21°C) by knocking down FGF21 gene. This observation, which is rather interesting, has the strong limitation of not controlling the side effects of FGF21 invalidation on a myriad of potential hormonal and metabolic factors which may react to FGF21 invalidation and constitute the actual direct actor/s of the compensatory process. It is obvious that, being FGF21 what is knocked out, it is the primary candidate to provide the compensatory mechanisms, but in the absence of additional experiments this cannot be stated in a conclusive manner without the possibility of relevant indirect effects. Addition of data using experimental approaches independent from FGF21-KO (perhaps reversion of FGF21 increase in UCP1-KO mice by in vivo immunization against FGF21 leading to normalization but not full suppression of FGF21, or genetic interventions on the FGF21 responsiveness machinery system) are necessary for the strong conclusive claim of the manuscript.

2. Moreover, the whole body knocking down of FGF21 makes the interpretation of data particularly complex and, in fact, a genetic intervention design that had used BAT (and beige) specific FGF21-KO (e.g. UCP1 promoter driven FGF21-KO) would have been more informative, specially considering that authors suggest (or insinuate) across the manuscript that BAT is the source of the high levels of FGF21 in blood from UCP1-KO mice which may account for the protection against obesity in UCP1-KO mice.

3. A weakness of the manuscript is that the actual mechanisms of action of the proposed FGF21-driven compensation in UCP1-KO mice are not clarified, and several obvious areas to be analyzed remain unexplored (which is in contrast with the otherwise extensive non-biased analytical procedures followed including lipidomics, microbiota analysis, RNAseq,.. from which no clear-cut mechanistic conclusion can be driven ultimately to explain the key phenotype of reversion of obesity protection in dKO mice). Considering the reported effects of FGF21 on "browning" of WAT (Fischer et al., 2012) and the signs of alterations in WAT transcriptome, non-UCP1-dependent mechanisms of energy expenditure potentially driven by FGF21 should be analyzed functionally in iWAT, i.e. the UCP1-independent, creatine-mediated pathways of energy expenditure in beige adipocytes, proposed by the Spiegelman group. Growing awareness of muscle-dependent mechanisms of diet-induced energy expenditure should perhaps deserve attention (Periasamy papers) considering recent publications relating FGF21 and muscle. Otherwise, are high FGF21 levels expected to moderate food consumption through central action, given previous research in the field?

4. The energy balance studies, which are key to identify the mechanisms underlying the reversion of obesity protection in dKO mice, are not fully clear. The lack of individual processing of data for energy intake versus energy outflow and therefore lack of standard deviations in means for the calculated ratios is a strong limitation, and clear-cut conclusions are hampered. Quantitative

considerations may be sound, but in the absence of a consistent statistical analysis of data, the current outcome of the manuscript in this regard is hardly conclusive.

Specific points:

The title should be modified to be more clear. In fact "a single endocrine factor" means "FGF21" and should be quoted directly as such, it would be more informative to readers. Moreover, the title is possibly somewhat overstated (see comments above) concerning the sole role of FGF21.

Reconsider the organization of the full Figures versus supplemental. For example, 8 panels in Fig 3 are devoted to show only a single significantly different result (water intake).

In Supplemental Table 1 there is a heading but not legend to define meaning of letters (a,b,...) for statistical significance.

Seminal papers on the effects of FGF21 on browning of WAT (Fisher et al. *Genes Dev* 2012) and FGF21 in relation to BAT activity and secretion (Hondares et al. *JBC* 2011) should be quoted and data discussed in relation to those previous observations.

Reference to models of impaired BAT function distinct from UCP1-KO in the last paragraph of Discussion is unclear. The lack of FGF21 increase in these models is speculative, isn't it? Are there data on actual measures of circulating FGF21 in those models?

Reviewer #2 (Remarks to the Author):

In their study Keipert et al., show that FGF21 is responsible for the obesity resistant phenotype of UCP1 deficient animals at ambient temperatures. Under high fat diet and mild cold stress conditions UCP-1 deficient mice massively upregulate FGF21 expression and serum concentrations. Eliminating FGF21 reverses the energy expending adipose tissue phenotype of UCP-1 deficient mice. Importantly, authors also show that solely measuring food intake and energy expenditure by indirect calorimetry does not provide enough information to explain an obesity resistant phenotype. They nicely cumulated all the small differences in bioenergetics parameters for their bioenergetics assessment. They claim that the "paradox" obesity resistance of UCP-1 ko mice" is due to an increase in futile cycling between lipid degradation and lipid synthesis.

General opinion:

Overall, the study is well designed and the experiments as well as the statistics are of high quality and adequate to the hypotheses. Every Figure legend contains the statistical tests used and the description of the error bars. Previous literature is appropriately cited. The abstract, introduction and conclusions are clear and appropriate.

Major points:

- Authors nicely show that the gene expression pattern in iWAT dramatically changes when FGF21 is deficient in Ucp-1-ko animals. However, and unfortunately, we are not provided with any mechanism of how FGF21 would cause these transcriptional changes (via PGC1a?) and whether any of these changes indeed translates into increased lipolytic or lipogenic activity.
- A prove for increased lipogenesis being responsible for energy expenditure in UCP-1 deficient animals is given by the decrease of OCR in the presence of Triacsin C. However, without showing that this effect is reversed in the double knock out adipose tissue, we don't know whether it depends on FGF21. Please provide the data including the double ko.
- If futile cycling of lipolysis/lipid synthesis is increased, there should be remodeling of iWAT towards a multilocular phenotype and the appearance of micro-lipid droplets within adipocytes. It would be desirable to include histological images showing the morphology of wt, UCP-1 ko, FGF21-ko and double-ko iWAT.
- All genotypes where raised at 30°C and then switched to 23°C and on HFD. First analyses, (including determination of FGF21 concentration in plasma) were performed 3 weeks after the switch, when body weight curves start to separate. Between week 0 and week 3 all genotypes gain similar weight. Does that indicate that it takes 3 weeks until FGF21 is increased in plasma; or that adipose tissue needs 3 weeks to be remodeled to increase energy expenditure? To understand this it would be good to show FGF21 plasma concentrations before putting the mice to 23°C and HFD, and compare it to the concentrations after 2 days, 1 week, 3 weeks, and 12 weeks at 23°C and HFD. Moreover, it should be shown whether FGF21 serum concentrations coincide with adipose tissue remodeling.
- FGF21 is known to reduce plasma TG levels (Schlein et al., Cell metab, 2016) and to affect adipocyte lipolysis. Is there any difference in plasma TG, fatty acid, or glycerol levels in the plasma of single and double ko mice on HFD and 23°C?

- FGF21 increases glucose uptake (Kharitonov et al. JCI 2005) and insulin sensitivity. Indeed, glucose and insulin concentrations are reduced in UCP-1 ko mice. However, they see no differences in glucose tolerance between the genotypes; My suggestion would be to perform ITT as it is the preferred method to measure insulin sensitivity.

- Beta klotho has been shown to be essential for FGF21 actions, and obesity is supposed to be an FGF21 resistant state with reduced Beta klotho expression (Fisher et al., Diabetes, 2010). Is there a difference in Beta klotho expression between wild-type and UCP1 deficient animals under HFD and mild cold conditions compared to 30°C chow?

- Some methods are missing in the method section like how bomb calorimetry was performed on feces samples or how food assimilation was calculated.

Minor points:

- It would be better for the reader to always show energy content using the same unit, (best would be kJ/d) and not switch between kJ and kcal (Metabolic rate figure 3 a is given in kcal/h and urine energy or fecal energy, fig 4 g and fig 5 m, is given in kJ/d).

- FGF21 is induced by ketogenic diet (Badman et al., Cell metab, 2007). Is its expression also induced by HFD feeding per se or only by the transfer of mice from 30°C to 23°C?

- Figure 2, body weights and body fat content of the different genotypes: Is there any explanation why mice at 30°C are less obese compared to mice at 23°C? From what is known, metabolic rate is decreased at 30°C, which would lead to fat accumulation.

Martina Schweiger

Reviewer #3 (Remarks to the Author):

The manuscript by Keipert et al is an interesting and extensive study that addresses the identification of the factors underlying the 'paradoxical' obesity resistance reported in UCP1-null mice at environmental conditions of mild cold stress (i.e., at housing room temperature). This group and others had previously reported that UCP1-KO mice show a high increase in FGF21 levels and FGF21 expression in brown adipose tissue (BAT) and white adipose tissue (WAT) (Keipert et al., 2015, ref 24 in the manuscript; Samms et al., Cell Rep 2015), suggestive of homeostatic compensatory

mechanisms for promotion of energy expenditure when the UCP1-mediated mechanisms are blunted. Here, using the UCP1/FGF21 double-KO mouse model, the authors demonstrate that the resistance to high-fat induced obesity in the UCP1-KO mice requires FGF21.

This novel observation is of great interest in the metabolic field because it contributes to the characterization of UCP1-independent pathways that cause resistance to obesity. However, the identification of the molecular mechanisms involved in the FGF21-mediated compensation in the UCP1-KO mice (and, therefore, absent in the UCP1/FGF21 double-KO) is not fully conclusive.

One point here is whether the effects of FGF21 are direct or indirect: other producing/target tissues of FGF21 may be involved (e.g., altering serum metabolites, induction of FGF21 expression in skeletal muscle, disturbed hepatic metabolism, paracrine action of FGF21 in BAT resulting in altered release of other batokines, ...). Considering that the global FGF21-KO is used, interpretation of data is complex and side effects of FGF21 invalidation cannot be ruled out.

The simplest explanation suggested by the authors is that over-expression of FGF21 in UCP1-defective BAT leads to increased circulating FGF21 that may induce a compensatory energy-burning mechanism/s in WAT:

- It has been reported that: FGF21 expression and secretion is induced in active BAT (Hondares et al., J Biol Chem 2011); FGF21 induces the browning of WAT (Fisher et al., Genes Dev 2012); pharmacological effects of FGF21 (weight loss, improved glucose homeostasis and plasma lipids, associated with increased energy expenditure) are also found in UCP1-KO mice (Veniant et al., Cell Metab 2015). This previous observations should be quoted and discussed in relation to present findings.

- An exhaustive number of experimental approaches have been used, involving metabolic and energy balance phenotyping, RNAseq analysis of BAT and iWAT, metabolomics, and microbiota analysis. However, in order to strengthen the conclusions, further characterization of iWAT would help: iWAT morphology to assess the degree of browning; assessment of FGF21 responsiveness machinery in iWAT; further identification of the alternative (UCP1-independent) pathways of energy loss in WAT. Although mRNA expression of some marker genes of alternative thermogenic pathways were found to be unaltered, a deeper characterization (functional if possible) of, e.g., the creatine-mediated system of energy dissipation (ref 38) would be of interest.

- Given transcriptional profiling data and some in vitro data in beige adipocytes, the authors suggest a lipid futile cycling promoting energy expenditure (simultaneous lipogenic and lipolytic metabolism). However, alterations in iWAT lipid metabolism (e.g., glyceroneogenesis) might also account for alterations in the inter-organ futile cycle between WAT and liver. In that sense, Fig 2F depicted that liver TG content is lower in UCP1-KO mice but not in the double-KO. How are the serum levels of TG, NEFA or glycerol?

- Regarding the GO enrichment analysis in iWAT (Fig.6e) and BAT (Fig.S3e). Could the comparison of UCP1-KO vs double-KO add more information to the specific altered enriched pathways explaining FGF21-mediated compensation in the UCP1-KO mice?

Other specific points:

- Regarding the Title, it should be modified to be more informative by adding FGF21.

- There is a discrepancy between Results, line 134 (nine weeks) and Fig.2 Leg, line 704 (eight weeks)

-Results, line 170, and Fig.4 Leg, line 724, please indicate that acyl-carnitines and amino acids are in urine.

-Statistical analysis and level of detail provided in Methods are adequate.

We are very grateful for the reviewers' helpful and constructive comments, which were critical for this revision process and helped to further improve the manuscript. We also thank the editors for their advice and the opportunity to address the concerns.

The comments by the reviewers have substantially strengthened the manuscript. Since the initial submission, we have performed a series of new experiments and analyses, including the investigation of other metabolically important tissues to address tissue-specificity of the observed metabolic effects. Notably, this includes next generation RNA sequencing of liver and skeletal muscle, the assessment of plasma parameters, adipose tissue morphology and network analysis. Thanks to the concerns of the reviewers, we have now a more complete understanding on tissues causing the metabolic phenotype, which are affected by endogenous FGF21.

We appreciate the reviewers' thoughts to elucidate all possible molecular alterations that FGF21 invokes on metabolism. However, we would also like to stress that this study represents the first work shedding light on the enigmatic factor that controls obesity resistance under room temperature in the UCP1 knockout mouse (Liu et al. 2003, Cannon and Nedergaard 2004); after more than 20 years of utilizing the UCP1 KO mouse as main tool to address brown fat function (ignoring the existence of paradoxical obesity resistance). Thus, we are confident that future research using the UCP1 KO mouse will consider our comprehensive characterization of the metabolic phenotype to elucidate all details that opens a new window to investigate UCP1-independent energy loss *in vivo*.

First, as suggested by the reviewer(s), we would specify the title of our manuscript, changing it to "Endogenous FGF21-signaling controls paradoxical obesity resistance of UCP1-deficient mice"

In the following, we address the reviewers' comments point-by-point:

Reviewers' comments:

Reviewer #1 (Remarks to the Author):

1. The core observation in the manuscript by Keipert et al. is the reversion of protection against weight gain in HFD-fed UCP1-KO versus wild-type mice (at environment conditions of mild cold such as 21°C) by knocking down FGF21 gene. This observation, which is rather interesting, has the strong limitation of not controlling the side effects of FGF21 invalidation on a myriad of potential hormonal and metabolic factors which may react to FGF21 invalidation and constitute the actual direct actor/s of the compensatory process. It is obvious that, being FGF21 what is knocked out, it is the primary candidate to provide the compensatory mechanisms, but in the absence of additional experiments this cannot be stated in a conclusive manner without the possibility of relevant indirect effects. Addition of data using experimental approaches independent from FGF21-KO (perhaps reversion of FGF21 increase in UCP1-KO mice by *in vivo* immunization against FGF21 leading to normalization but not full suppression of FGF21, or genetic interventions on the FGF21 responsiveness machinery system) are necessary for the strong conclusive claim of the manuscript.

RESPONSE: We thank the reviewer for his/her thoughtful comments and ideas. We fully agree that metabolism is a multi-faceted process that is adjusted by complex signaling and feedback mechanisms. We agree with the reviewer, that FGF21 is the

primary candidate and from our joint point of view, the master regulator of the UCP1 KO invoked obesity protection. We agree that indirect effects downstream of FGF21 may play a role, thus we removed passages, which claim that our study fully solves how FGF21 alters tissue-specific signaling. However, the reviewer also rightfully suggested citing seminal papers in the field (we here apologize for not having them included earlier). Notably, these papers on the effects of FGF21 on browning of WAT (Fisher et al. 2012, PMID: 22302939; Ost et al. 2016, PMID: 26909316) and FGF21 in relation to BAT activity and secretion (Hondares et al. 2011, PMID: 21317437) provide strong evidence that FGF21 signaling and WAT browning are directly related. We hope that the reviewer agrees that the identification of pivotal factor for obesity resistance in UCP1 KO mice is a major step forward not only understanding this phenomenon after such a long time, but also is important knowledge for other studies using the UCP1 KO mouse below thermoneutral conditions.

After our initial finding involving the generation of dKO mice, we invested major efforts with multi-tissue transcriptomics, comprehensive mouse metabolic phenotyping (including challenging urine bioenergetics) and in vitro experimentation, to pinpoint responsible tissue sites and exclude metabolic factors and experimental approaches (e.g. in vitro adipocytes do not resemble bioenergetics differences). In the revision we further analyzed our transcriptomics data whether batokines other than FGF21 may play a role for obesity resistance (new Figure 5F). While we are planning to further investigate the molecular mechanisms with new animal ethics protocols, such as immunization and further genetic models, we feel that the current discovery and characterization justifies dissemination to the research community.

While the genetic ablation of FGF21 to show the systemic role of endogenous FGF21 could always be complemented with additional experiments, alternative approaches may also come with caveats (e.g. off-targets of in vivo immunization). To support the specific metabolic importance of white fat browning, we have now generated RNA sequencing of two other important metabolic tissues, which can also be source and target tissues of FGF21, skeletal muscle and the liver. These data demonstrate that in these organs, minor UCP1 and FGF21-dependent changes occur as compared to white adipose tissue, supporting the conclusion that white fat browning is a major metabolic effector of endogenous FGF21 signaling. To fairly discuss our findings, we tempered our conclusions and keep the reviewer's ideas in mind for future studies.

We added to the discussion: *“Seminal findings by others demonstrated FGF21's potency to directly brown WAT (Fisher et al. 2012, Ost et al 2016), but whether FGF21 impacts browning also indirectly in the UCP1 KO mouse, remains to be determined.”*

We present new results, showing minor changes in liver and skeletal muscle (new Figures 5 c, d), no association of typical batokines to FGF21-responsiveness and obesity resistance (new Figure 5f).

2. Moreover, the whole body knocking down of FGF21 makes the interpretation of data particularly complex and, in fact, a genetic intervention design that had used BAT (and beige) specific FGF21-KO (e.g. UCP1 promoter driven FGF21-KO) would had been more informative, specially considering that authors suggest (or insinuate) across the manuscript that BAT is the source of the high levels of FGF21 in blood from UCP1-KO mice which may account for the protection against obesity in UCP1-KO mice.

RESPONSE: We think that in a series of papers, we have established that in the absence of UCP1, brown adipose tissue becomes a significant source of FGF21

(Keipert et al. 2015, PMID: 26137441; Keipert et al. 2017, PMID: 28768181). As FGF21 is a stress-responsive gene usually associated to the liver, it is not enhanced in our husbandry conditions of thermoneutrality. We carefully checked in all of our studies liver FGF21 levels to rule out its involvement. We observe induction of FGF21 in UCP1 KO mice only in adipose tissue upon cold stress. As the reviewer may appreciate, the generation of double knockouts is a mouse number/breeding intensive operation, and it would almost exhaust our capacities to additionally crossbreed adipose-specific drivers into this double knockout. We agree, that if we had observed genotypic differences of FGF21 induction in other tissues, we would have had to generate tissue-specific knockouts. We are evaluating the possibility of generating UCP1-FGF21 double-knockouts with CRISPR technologies, but these efforts are preliminary and beyond the timeframe of a reasonable major revision.

3. A weakness of the manuscript is that the actual mechanisms of action of the proposed FGF21-driven compensation in UCP1-KO mice are not clarified, and several obvious areas to be analyzed remain unexplored (which is in contrast with the otherwise extensive non-biased analytical procedures followed including lipidomics, microbiota analysis, RNAseq,.. from which no clear-cut mechanistic conclusion can be driven ultimately to explain the key phenotype of reversion of obesity protection in dKO mice). Considering the reported effects of FGF21 on "browning" of WAT (Fischer et al., 2012) and the signs of alterations in WAT transcriptome, non-UCP1-dependent mechanisms of energy expenditure potentially driven by FGF21 should be analyzed functionally in iWAT, i.e. the UCP1-independent, creatine-mediated pathways of energy expenditure in beige adipocytes, proposed by the Spiegelman group. Growing awareness of muscle-dependent mechanisms of diet-induced energy expenditure should perhaps deserve attention (Periasamy papers) considering recent publications relating FGF21 and muscle. Otherwise, are high FGF21 levels expected to moderate food consumption through central action, given previous research in the field?

RESPONSE: We agree that it is a major weakness that the actual mechanism of action surrounding FGF21-driven weight loss has not been clarified in the vast body of laboratories investigating the metabolic role of FGF21, and we also cannot provide the ultimate answer. In this manuscript, concerning energy loss, we have investigated several areas that are logic from the bioenergetics point of view and provide these unbiased data. We feel that these results, although they do not always show significant differences, are important for the research community, by excluding possible physiological mechanisms. Furthermore, collectively, these data are coherently characterizing the metabolic phenotype of UCP1-independent obesity resistance, and are partially in line with previous attempts (Liu et al. 2003). Furthermore, we approached the bioenergetic consequences in vitro using adipocytes of all genotypes. However, the molecular phenotype in vivo could not be mimicked identically in the vitro situation, and thus, make the adipocytes unfortunately an unfavorable system. Pertaining to alternative mechanisms, we searched for evidence in our transcriptomic data set (Fig. 5g), including creatine futile cycling, as also highlighted by the reviewer. However, the Spiegelman group found differentially regulated genes of creatine futile cycling in beige fat of cold-acclimated mice, while we are using mice at room temperature, fed with high-fat diets. Thus, these conditions do not appear comparable. In our mice, the transcriptional setup did not provide evidence for differentially regulated creatine futile cycling, as found for the cold-acclimated mice (e.g. Ckmt1 expression was below detection levels). In our mice, we

also do not see striking transcriptional regulation of other suggested pathways. Thus, we did not follow up initially on these mechanisms, as they were, in contrast to these seminal papers by others, not supported by transcriptional recruitment (Figure 5g). Non-shivering thermogenesis in muscle may be considered but the field of muscle non-shivering thermogenesis remains controversial (e.g. Campbell and Dicke 2018, PMID: 29962960). In the light of minor changes in our muscle transcriptome (in total only 15 DEGs when comparing UCP1 KO with dKO mice, see Fig. 5e), it would rather weaken our study to speculate on the existence of muscle NST.

The issue whether FGF21 affects food-consumption, is controversial in the field. In WT mice increased levels of FGF21 are always associated with increased energy expenditure and no difference in food intake. For example, the reduction of food intake was not seen during FGF21 gene therapy (V Jimenez et al. 2018, PMID: 29987000), during transgenic overexpression of FGF21 (Zhang et al, 2012 PMID: 23066506), or during exogenous FGF21 administration (Coskun et al., 2008, PMID: 18687777). Interestingly, in contrast to these rodent studies, the FGF21 agonist PF-05231023 reduced body weight in monkeys by decreasing food intake suggesting species specific action of FGF21 (Talukdar et al., 2016, PMID: 26959184). While an effect on food intake could be found in obese UCP1 KO mice using supraphysiological doses of exogenous FGF21 (Samms et al., 2015, PMID: 25956583), the pair-feeding experiment revealed that reduced food intake does not completely explain the body weight reduction. However, our data reveal no significant impact on food consumption by physiological relevant levels of endogenous FGF21. Notably, however, the recently published effect on drinking behavior is seen in our metabolic phenotyping, supporting the notion that the endogenous levels of FGF21 have metabolic impact. We thank the reviewer for the important notion of food intake that we also included in our discussion.

Discussion: 'Our data reveal no significant impact of endogenous FGF21 on food consumption. In contrast, high doses of exogenous FGF21 reduce food intake of obese UCP1 KO mice (Samms et al., 2015, PMID: 25956583). However, pair-feeding experiments also reveal that this only partially explains the body weight reduction. In WT mice, increased levels of FGF21 are usually associated with increased energy expenditure but no differences in food intake, as seen during FGF21 gene therapy (V Jimenez et al. 2018, PMID: 29987000), by transgenic overexpression of FGF21 (Zhang et al, 2017 PMID: 23066506) and exogenous FGF21 administration (Coskun et al., 2008, PMID: 18687777).'

4. The energy balance studies, which are key to identify the mechanisms underlying the reversion of obesity protection in dKO mice, are not fully clear. The lack of individual processing of data for energy intake versus energy outflow and therefore lack of standard deviations in means for the calculated ratios is a strong limitation, and clear-cut conclusions are hampered. Quantitative considerations may be sound, but in the absence of a consistent statistical analysis of data, the current outcome of the manuscript in this regard is hardly conclusive.

RESPONSE: Reviewer's point well taken. We agree that our attempt to cumulatively analyze the metabolic phenotyping data to explain minor differences developing into severe obesity, do not allow clear-cut conclusions. However, approaching these considerations of minor changes appeared important for the understanding of obesity development, which is indeed well known to clinicians working on human obesity.

Thus, to keep these thoughts alive but to temper our conclusions, we moved these semi-quantitative considerations from the results to the discussion of the data and shortened it considerably. We hope that the reviewer agrees that it is noteworthy reporting this without claiming clear-cut conclusions.

Specific points:

The title should be modified to be more clear. In fact "a single endocrine factor" means "FGF21" and should be quoted directly as such, it would be more informative to readers. Moreover, the title is possibly somewhat overstated (see comments above) concerning the sole role of FGF21.

RESPONSE: We agree with the reviewer and changed the title to “Endogenous FGF21-signaling controls paradoxical obesity resistance of UCP1-deficient mice”.

Reconsider the organization of the full Figures versus supplemental. For example, 8 panels in Fig 3 are devoted to show only a single significantly different result (water intake).

RESPONSE: We agree with the reviewer that some of the figures can be re-arranged. We still think that not only significantly changing parameters in mouse metabolic phenotyping are important but also not changing parameters to exclude causal effectors of obesity. However, we compromised under the umbrella ‘mouse metabolic phenotyping’ not only the energy flow ‘in’ (food/water intake) and ‘between’ (MR, activity, Tb, RER), but also ‘out’ (microbiota, feces energy and food assimilation). Thus, we omitted the previous Figure 5 and moved parts to the supplement.

In Supplemental Table 1 there is a heading but not legend to define meaning of letters (a,b,...) for statistical significance.

RESPONSE: We thank the reviewer for this notion and added this.

Seminal papers on the effects of FGF21 on browning of WAT (Fisher et al. Genes Dev 2012) and FGF21 in relation to BAT activity and secretion (Hondares et al. JBC 2011) should be quoted and data discussed in relation to those previous observations.

RESPONSE: We included these references, such as Fisher et al., as mentioned above, and Hondares in “In previous studies, others showed that BAT becomes a source of FGF21 in the cold (Hondares et al. 2011, PMID: 21317437), which is further potentiated in UCP1 KO mice (Keipert et al. 2015, PMID: 26137441).”

Reference to models of impaired BAT function distinct from UCP1-KO in the last paragraph of Discussion is unclear. The lack of FGF21 increase in these models is speculative, isn't it? Are there data on actual measures of circulating FGF21 in those models?

RESPONSE: We agree that this part of the discussion is an educated guess for further investigations and thus, we changed the wording to flag it as confounded speculation.

Reviewer #2 (Remarks to the Author):

In their study Keipert et al., show that FGF21 is responsible for the obesity resistant phenotype of UCP1 deficient animals at ambient temperatures. Under high fat diet and mild cold stress conditions, UCP-1 deficient mice massively upregulate FGF21 expression and serum concentrations. Eliminating FGF21 reverses the energy expending adipose tissue phenotype of UCP-1 deficient mice. Importantly, authors also show that solely measuring food intake and energy expenditure by indirect calorimetry does not provide enough information to explain an obesity resistant phenotype. They nicely cumulated all the small differences in bioenergetics parameters for their bioenergetics assessment. They claim that the “paradox” obesity resistance of UCP-1 ko mice” is due to an increase in futile cycling between lipid degradation and lipid synthesis.

General opinion:

Overall, the study is well designed and the experiments as well as the statistics are of high quality and adequate to the hypotheses. Every Figure legend contains the statistical tests used and the description of the error bars. Previous literature is appropriately cited. The abstract, introduction and conclusions are clear and appropriate.

RESPONSE: We thank the reviewer for a balanced and positive evaluation of our work.

Major points:

- Authors nicely show that the gene expression pattern in iWAT dramatically changes when FGF21 is deficient in Ucp-1-ko animals. However, and unfortunately, we are not provided with any mechanism of how FGF21 would cause these transcriptional changes (via PGC1a?) and whether any of these changes indeed translates into increased lipolytic or lipogenic activity.

RESPONSE: We agree with the reviewer that we do not provide causal experimentation on transcriptional control via FGF21. In our in vivo study, it is hard to dissect unambiguously which multiple processes feed into the control of enhanced lipid gene transcription. Indeed, experts judge FGF21-dependent signaling as ‘extremely complicated’ (Tezze et al., 2019, PMID: 31057418). We are relying on seminal previous papers that focus on the intracellular signaling cascades mediated by FGFRs in combination with beta klotho (Ogawa et al., 2007 PMID: 17452648). We cite this paper while discussing potential pathways deduced from our transcriptomic analysis (*‘...FGF21-dependent intracellular signaling is mediated by FGFRs in combination with beta klotho (Ogawa et al. 2007, PMID: 17452648), but other unknown factors may contribute. While we did not investigate novel factors of intracellular FGF21 signaling, the genetic network underlying browning and metabolic consequences in iWAT were addressed in silico to get further insights into potential pathways.’*)

To address the metabolic consequences of the genetic network underlying FGF21-dependent browning, we performed several bioinformatic analyses, including the comparison our of transcriptomic data in UCP1 KO iWAT with the recently published molecular signature of browning (Cheng et al. 2018, PMID: 29874595), that is based on meta-analysis of more than 100 data sets. It transpires from this unbiased, systemic analysis, that the typical browning signature is induced in UCP1 KO mice, controlled by PPAR alpha and PGC1 alpha, which were more pronouncedly increased than

Nr4A1, an adrenergically responsive gene (see supplemental figure 3c). The metabolic pathway analysis applying these transcription factors on highly significant 244 DEGs (UCP1 KO vs dKO; adjusted pvalue < 0.001) maps potential routes of transcriptional control that induce lipid metabolism (see supplemental figure 3d). The regulation analysis is also found in supplemental figure 3, suggesting PPAR alpha and PGC1 alpha as controlling factors of lipid associating genes such as Cpt1b and PDK4. We refer to this analysis in the discussion: *'The recently published molecular browning signature and its regulation, that is based on meta-analysis of more than 100 published data sets (Cheng et al. 2018, PMID: 29874595), has been adopted to our data for prediction of regulatory pathways. This analysis reveals the molecular induction of browning, controlled by PPAR alpha and PGC1 alpha, which were more pronouncedly increased than NR4A1, an adrenergically responsive gene (see supplemental figure 3d). Mapping these transcription factors on highly significant DEGs of iWAT in UCP1 KO mice elucidates routes for the induction of lipid metabolism genes, such as CPT1b and PDK4. This is further supported by an unbiased pathway analysis of all transcriptomic data, revealing the FGF21-dependent enrichment of lipid and oxidative metabolism only in iWAT, including the typical browning genes (Cheng et al. 2018, PMID: 29874595)'*

- A prove for increased lipogenesis being responsible for energy expenditure in UCP-1 deficient animals is given by the decrease of OCR in the presence of Triacsin C. However, without showing that this effect is reversed in the double knock out adipose tissue, we don't know whether it depends on FGF21. Please provide the data including the double ko.

RESPONSE: Although we support the route to link lipid metabolism with increased energy expenditure, based on our and data of others (e.g. Rohm et al. 2016, PMID: 27571348; Schlein et al. 2016, PMID: 26853749), and clinical studies (e.g. by the Arner group, supporting imbalanced lipid metabolism as driver for human obesity), we omitted our in vitro results as too preliminary. First, we cannot fully mimic the FGF21-dependent browning conditions of the genotypes in vitro, and second, we became aware that Triacsin C is not solely a "lipogenesis inhibitor" but besides other off-target effects, the inhibition of fatty acid activation is confounding, which would, in the bioenergetics analysis, simultaneously inhibit lipolytic processes. However, we think that our transcriptomic data would still allow us to conclude induced lipid metabolism in WAT.

- If futile cycling of lipolysis/lipid synthesis is increased, there should be remodeling of iWAT towards a multilocular phenotype and the appearance of micro-lipid droplets within adipocytes. It would be desirable to include histological images showing the morphology of wt, UCP-1 ko, FGF21-ko and double-ko iWAT.

RESPONSE: We agree with the reviewer that histological images improve our study. Indeed we detect multilocular lipid droplets only in iWAT of UCP1 KO mice but not in the other genotypes. We added the histological pictures in supplemental figure 4a.

- All genotypes were raised at 30°C and then switched to 23°C and on HFD. First analyses, (including determination of FGF21 concentration in plasma) were performed 3 weeks after the switch, when body weight curves start to separate. Between week 0 and week 3 all genotypes gain similar weight. Does that indicate that it takes 3 weeks until FGF21 is

increased in plasma; or that adipose tissue needs 3 weeks to be remodeled to increase energy expenditure? To understand this it would be good to show FGF21 plasma concentrations before putting the mice to 23°C and HFD, and compare it to the concentrations after 2 days, 1 week, 3 weeks, and 12 weeks at 23°C and HFD. Moreover, it should be shown whether FGF21 serum concentrations coincide with adipose tissue remodeling.

RESPONSE: We agree that the time-resolution is very informative but we have simply not included these detailed experiments in our animal ethics protocols. We do not want to speculate too much but FGF21 serum levels of animals kept at 30°C on a chow diet (in this manuscript: time point zero) are not different between WT and UCP1 KO mice (overall very low levels). In a previous study, we find significantly elevated FGF21 levels only after 2 weeks of room temperature in UCP1 KO mice on chow (Keipert et al. 2015, PMID: 26137441). Samms et al. (2015, PMID: 25956583) treated obese mice with a very high dose of FGF21 (1 mg/kg/day) and could detect differences in „browning“ related genes of iWAT only after 7 days. Collectively, these studies strongly suggest that FGF21 induced browning is not an acute response in vivo.

• FGF21 is known to reduce plasma TG levels (Schlein et al., Cell metab, 2016) and to affect adipocyte lipolysis. Is there any difference in plasma TG, fatty acid, or glycerol levels in the plasma of single and double ko mice on HFD and 23°C?

Response: We measured serum TG and NEFA levels in our mice cohorts and added the results in Fig 2i and Fig2m.

• FGF21 increases glucose uptake (Kharitonov et al. JCI 2005) and insulin sensitivity. Indeed, glucose and insulin concentrations are reduced in UCP-1 ko mice. However, they see no differences in glucose tolerance between the genotypes; My suggestion would be to perform ITT as it is the preferred method to measure insulin sensitivity.

Response: Indeed all genotypes show the same glucose tolerance and no signs of diabetes but the reduced insulin levels in UCP1 KO mice point to improved insulin sensitivity after eight weeks of HFD feeding. We now calculated and included in Figure 2 the HOMA-IR, which gives another well accepted parameter for differences in insulin resistance, again showing improved insulin sensitivity in UCP1 KO mice compared to dKO. An additional ITT would indeed be helpful as well, but generating a new cohort of adult double KO mice would take at least another year as it involves 3-generation cycles of mouse breeding.

• Beta klotho has been shown to be essential for FGF21actions, and obesity is supposed to be an FGF21 resistant state with reduced Beta klotho expression (Fisher et al., Diabetes, 2010). Is there a difference in Beta klotho expression between wild-type and UCP1 deficient animals under HFD and mild cold conditions compared to 30°C chow?

Response: We measured KLB gene expression in iWAT of mice fed HFD for 3 and 12 wks at room temperature and compared those to mice fed HFD for 12 wks at thermoneutrality (see figure below). Overall KLB gene expression was reduced after long term HFD feeding (which is in line with published data; Markan et al. 2017, PMID: 28580290), independent of ambient temperature. We could observe significant

differences between WT and UCP1 KO mice kept at room temperature after long-term high fat diet, which hints towards improved FGF21 sensitivity in iWAT of UCP1 KO mice. However, the impact of KLB expression in iWAT on the phenomenon of “FGF21 resistance” during obesity is still a matter of debate. While the overexpression of KLB in iWAT, but not in liver, leads to protection against diet-induced obesity (DIO) (Samms et al. 2016, PMID: 26901091) in one study, another paper reports on the maintenance of KLB in iWAT during HFD feeding that does not protect from DIO (Markan et al. 2017, PMID: 28580290). We agree with the reviewer that studying the impact of FGF21 resistance or improved sensitivity to protect against DIO is of great interest and we will focus on that topic in future studies.

[Redacted]

- Some methods are missing in the method section like how bomb calorimetry was performed on feces samples or how food assimilation was calculated.

Response: Thank you for bringing this to our attention. We added feces bomb calorimetry to the methods section.

Minor points:

- It would be better for the reader to always show energy content using the same unit, (best would be kJ/d) and not switch between kJ and kcal (Metabolic rate figure 3 a is given in kcal/h and urine energy or fecal energy, fig 4 g and fig 5 m, is given in kJ/d).

Response: We agree with the reviewer and changed the unit for metabolic rate to kJ/d throughout the manuscript (see Figure 3a, 3b, S2b and Supplemental table 2).

- FGF21 is induced by ketogenic diet (Badman et al., Cell metab, 2007). Is its expression also induced by HFD feeding per se or only by the transfer of mice from 30°C to 23°C?

Response: We measured FGF21 gene expression in BAT and iWAT of mice fed HFD for 3 and 12 wks at room temperature and compared those to mice fed HFD for 12 wks at thermoneutrality (see figure below). FGF21 is only induced in adipose tissue of UCP1 KO mice on high fat diet in combination with mild cold exposure. UCP1 KO mice fed a HFD kept at the ambient temperature of 30°C show no detectable FGF21 expression in BAT or iWAT. We thank the reviewer for this notion and included these data in supplemental figure 1d.

Figure: FGF21 expression in (A) BAT and (B) iWAT of WT, FGF21 KO, UCP1 KO and dKO mice fed a high fat diet for 12 weeks (at room temperature-RT) or 12 wks kept at thermoneutrality (30°C). Data are presented as mean + SEM. (n = 6-8 per group).

• Figure 2, body weights and body fat content of the different genotypes: Is there any explanation why mice at 30°C are less obese compared to mice at 23°C? From what is known, metabolic rate is decreased at 30°C, which would lead to fat accumulation.

Response: We have also noticed this but have not followed up, as we focused on the role of UCP1 and FGF21. We agree with the reviewer that reduced RMR would contribute to obesity progression, but mice at thermoneutrality also eat significantly less, counter-acting obesity. However, UCP1 KO compared to WT mice at thermoneutrality were slightly more prone to DIO, coherent with previously published observations (PMID: 19187776).

Reviewer #3 (Remarks to the Author):

The manuscript by Keipert et al is an interesting and extensive study that addresses the identification of the factors underlying the 'paradoxical' obesity resistance reported in UCP1-null mice at environmental conditions of mild cold stress (i.e., at housing room temperature). This group and others had previously reported that UCP1-KO mice show a high increase in FGF21 levels and FGF21 expression in brown adipose tissue (BAT) and white adipose tissue (WAT) (Keipert et al., 2015, ref 24 in the manuscript; Samms et al., Cell Rep 2015), suggestive of homeostatic compensatory mechanisms for promotion of energy expenditure when the UCP1-mediated mechanisms are blunted. Here, using the UCP1/FGF21 double-KO mouse model, the authors demonstrate that the resistance to high-fat induced obesity in the UCP1-KO mice requires FGF21. This novel observation is of great interest in the metabolic field because it contributes to the characterization of UCP1-independent pathways that cause resistance to obesity. However, the identification of the molecular mechanisms involved in the FGF21-mediated compensation in the UCP1-KO mice (and, therefore, absent in the UCP1/FGF21 double-KO) is not fully conclusive. One point here is whether the effects of FGF21 are direct or indirect: other producing/target tissues of FGF21 may be involved (e.g., altering serum metabolites, induction of FGF21 expression in skeletal muscle, disturbed hepatic metabolism, paracrine action of FGF21 in BAT resulting in altered release of other batokines, ...). Considering that the global FGF21-KO is used, interpretation of data is complex and side effects of FGF21 invalidation cannot be ruled out.

Response: We thank the reviewer for agreeing with the main message of this study, claiming that the resistance to high-fat induced obesity in the UCP1-KO mice requires FGF21. The mysterious ‘anti-obesity’ factor, which is responsible for the “paradoxical” DIO resistance of UCP1 KO mice, has been an unresolved question for more than a decade and its identification as FGF21 should be of great interest to the metabolic research field that frequently uses the UCP1 KO mouse. Beyond the identification of this factor, we did our best to pinpoint responsible tissue-sites and potential mechanisms by fusing comprehensive metabolic phenotyping, biochemical and molecular analyses. As systemic metabolism in vivo is a multi-faceted process that is adjusted by complex signaling and feedback mechanisms, we cannot formally exclude whether some of the metabolic effects of BAT secreted FGF21 are direct or indirect, or whether they require secondary factors.

We admit that redoing this study with tissue-specific knockouts, breeding them to double knockouts with UCP1 KO mice, covering all genotypes, would not only exceed the timeframe, but also requires major mouse numbers that could be beyond a reasonable major revision. To characterize the source of obesity resistance, we invested major efforts to generate robust data on global transcriptomics by RNAseq analysis of all four genotypes in four major tissues (now also including liver and muscle) to strengthen iWAT as the target tissue of FGF21 action. These new RNA seq data on liver and muscle demonstrate only minor UCP1 and FGF21-dependent changes. Overall we could only detect 6 DEGs in liver and 15 DEGs in muscle when comparing UCP1 KO mice with dKO mice (see also Figure 5e), whereas more than 500 genes are differentially expressed in iWAT supporting our conclusion that white fat browning is a major metabolic effector of endogenous FGF21 signaling.

We now also included gene expression analysis of known batokines, to evaluate if other batokines are compensating the lack of FGF21 in dKO mice - which is not the case (see Figure 5f). No batokine is higher expressed in dKO compared to UCP1 KO mice, suggesting no compensatory mechanism. Impressively, FGF21 is the only batokine induced only in UCP1 KO mice compared to the other genotypes. A few batokines associated with thermoregulation (f.e. *Bmp8b*, *Epdr1*) are upregulated in both UCP1 KO and dKO mice, compared to WT mice thus they could not be responsible for the obesity resistance phenotype.

The simplest explanation suggested by the authors is that over-expression of FGF21 in UCP1-defective BAT leads to increased circulating FGF21 that may induce a compensatory energy-burning mechanism/s in WAT:- It has been reported that: FGF21 expression and secretion is induced in active BAT (Hondares et al., J Biol Chem 2011); FGF21 induces the browning of WAT (Fisher et al., Genes Dev 2012); pharmacological effects of FGF21 (weight loss, improved glucose homeostasis and plasma lipids, associated with increased energy expenditure) are also found in UCP1-KO mice (Veniant et al., Cell Metab 2015). This previous observations should be quoted and discussed in relation to present findings.

Response: We thank the reviewer for the suggested references and apologize that we have not cited and discussed this important findings in the first submission. The publications are now included in the revised manuscript.

‘In previous studies, others showed that BAT becomes a source of FGF21 in the cold (Hondares et al. 2011),...’

‘Seminal findings by others demonstrated FGF21’s potency to directly brown WAT (Fisher et al. 2012; Ost et al. 2016) but whether FGF21 impacts browning also indirectly in the UCP1 KO mouse, remains to be determined.’

‘These observations are coherent with the pharmacology of exogenous FGF21 that does not require UCP1-dependent thermogenesis for beneficial metabolic effects during obesity (Veniant et al. 2015; Samms et al. 2015).’

- An exhaustive number of experimental approaches have been used, involving metabolic and energy balance phenotyping, RNAseq analysis of BAT and iWAT, metabolomics, and microbiota analysis. However, in order to strengthen the conclusions, further characterization of iWAT would help: iWAT morphology to assess the degree of browning; assessment of FGF21 responsiveness machinery in iWAT; further identification of the alternative (UCP1-independent) pathways of energy loss in WAT. Although mRNA expression of some marker genes of alternative thermogenic pathways were found to be unaltered, a deeper characterization (functional if possible) of, e.g., the creatine-mediated system of energy dissipation (ref 38) would be of interest.

Response: We thank the reviewer for these valuable suggestions. We have now included histological images of iWAT (see supplemental figure 4a). Indeed we detect multilocular lipid droplets only in iWAT of UCP1 KO mice and not in the other genotypes proposing an increased “browning” of iWAT in UCP1 KO mice. Furthermore we compared our transcriptomic data with the recently published browning signature (Cheng et al. 2018, PMID: 29874595), that is based on meta-analysis of more than 100 data sets (see supplemental figure 3c). It transpires from this unbiased, systemic analysis, that the typical browning signature is induced in UCP1 KO mice, controlled by more pronouncedly induced PPAR alpha and PGC1 alpha, rather than by Nr4A1, an adrenergically responsive gene.

We also generated for another pharmacological study preliminary data on FGF21 responsiveness in high fat diet fed WT compared to UCP1 KO mice, suggesting a sustained FGF21 sensitivity in UCP1 KO mice after long term high fat diet feeding (Figure for reviewer only, below). Upon FGF21 administration, UCP1 KO mice lost weight, whereas WT appeared to be resistant in the initial phase of the treatment and, only in UCP1 KO mice, FGF21 treatment induced FGF21-responsive genes (Dio2, Cidea) in iWAT, further suggesting iWAT as the key tissue regulating metabolic improvements in UCP1 KO mice.

[Redacted]

For creatine cycling, we are currently establishing its evaluation in the XF96 Seahorse (originally published for the XF24), which does not appear to be an easy task, as creatine cycling with sub-saturating concentrations of creatine is currently not reproducible in our machinery setup. Thus, we would like to remain with the transcriptional data, which does not provide evidence for differentially regulated creatine futile cycling, as found for the cold-acclimated mice (e.g. Ckmt1 expression was below detection levels). Notably, the Spiegelman group found differentially regulated genes of creatine futile cycling in beige fat of cold-acclimated mice, while we are using mice at room temperature, fed with high-fat diets. Thus, these conditions do not appear comparable. In our mice, we also do not see striking transcriptional regulation of other suggested pathways. Thus, we did not follow up on these mechanisms for this paper, as they were, in contrast to seminal papers by others, not supported by transcriptional recruitment (Figure 5g).

- Given transcriptional profiling data and some in vitro data in beige adipocytes, the authors suggest a lipid futile cycling promoting energy expenditure (simultaneous lipogenic and lipolytic metabolism). However, alterations in iWAT lipid metabolism (e.g., glyceroneogenesis) might also account for alterations in the inter-organ futile cycle between WAT and liver. In that sense, Fig 2F depicted that liver TG content is lower in UCP1-KO mice but not in the double-KO. How are the serum levels of TG, NEFA or glycerol?

Response: We think that the liver triglyceride content depends primarily on the obesity phenotype, as after 3 wks HFD feeding (where already remodeling of iWAT is present in UCP1 KO mice) no differences in liver TG could be observed between UCP1 KO and dKO mice (see figure 2f). Now, we can also provide transcriptomic insights for the liver and the muscle, where not much difference is seen. We have now also performed and included serum analysis of Nefa and triglycerides (see Figure 2k and 2m).

- Regarding the GO enrichment analysis in iWAT (Fig.6e) and BAT (Fig.S3e). Could the comparison of UCP1-KO vs double-KO add more information to the specific altered enriched pathways explaining FGF21-mediated compensation in the UCP1-KO mice?

We are very thankful for the reviewer's suggestion to compare UCP1 KO mice directly with dKO mice. The results of this analysis clearly show that the main target tissue of endogenous FGF21 signaling is iWAT (see also Figure 5e and 5h).

Other specific points:

-Regarding the Title, it should be modified to be more informative by adding FGF21.

Response: We agree with the reviewer and changed the title to “*Endogenous FGF21-signaling controls paradoxical obesity resistance of UCP1-deficient mice*”

-There is a discrepancy between Results, line 134 (nine weeks) and Fig.2 Leg, line 704 (eight weeks)

Response: Thank you for bringing this to our attention, the GTT was performed after eight weeks of HFD feeding. We corrected the figure legend accordingly.

-Results, line 170, and Fig.4 Leg, line 724, please indicate that acyl-carnitines and amino acids are in urine.

Response: Added

-Statistical analysis and level of detail provided in Methods are adequate.

REVIEWERS' COMMENTS:

Reviewer #1 (Remarks to the Author):

The revised manuscript by Keipert et al. includes several improvements relative to the first version. The way data are presented and inclusion of more balanced statements and Discussion improved the manuscript. Some of my general concerns, however remain. Regarding point 1, concerning the specificity of the effects: the authors provide, for their double KO models: a) negative data on expression (mRNA levels, not actual circulating levels) of potential brown adipokines other than FGF21, b) negative transcriptomics data on main effects in liver and skeletal muscle. These data support to some extent the specificity of FGF21 but are not fully conclusive, as stated appropriately in the new paragraph in the Discussion stating this limitation. In my opinion, however, the main remaining weakness remaining in the revised manuscript concerns the absence of a functional explanation on how the browning of WAT, in the absence of UCP1-mediated energy expenditure mechanisms, may account for protection against obesity remain (Point 3 in the previous report). This is a key and relevant point, because general assumptions in the field are that WAT browning increase energy expenditure because: a) beige adipocytes possess UCP1-dependent energy expenditure, and/or b) additional mechanisms other than UCP1 which account for energy expenditure in beige adipocytes (Spiegelman group contributions, a much less consolidated view, or other possibilities). The observations in this manuscript are important because are strongly supportive of the relevance of b), but the lack of direct assessment of what is going on in WAT in bioenergetic terms is a weakness. In their approaches at this point (and other sections of the manuscript), the authors explore these issues mostly through by omics (mostly transcriptomics) data which provide steady-state data that suggest the involvement of pathways and physiological processes, but do not involve an actual exploration of function which would require direct, bioenergetically meaningful, experimentation. Moreover, the very moderate effects found when energy balance data are processed remain puzzling. Moving the data to discussion may be appropriate, but clear conclusions on whole body energy balance are nuclear to a study focused to mechanisms of obesity protection. In summary, the authors provide in their study a relevant information regarding the identification of what leads to protection against obesity in the UCP1-KO mouse model: the induction of FGF21 and resulting browning of WAT, but the mystery remains on how WAT browning in the absence of UCP1 may account for obesity protection. Without underestimating the very valuable information provided by the current manuscript, it's a pity this remains unclear, as this is a very relevant biological point.

Reviewer #2 (Remarks to the Author):

In their study Keipert et al., show that FGF21 is responsible for the obesity resistant phenotype of UCP1 deficient animals at ambient temperatures.

Overall, Keipert et al., answered the major and minor concerns I had, to my full satisfaction.

In detail:

Major points:

- Authors nicely show that the gene expression pattern in iWAT dramatically changes when FGF21 is deficient in Ucp-1-ko animals. However, and unfortunately, we are not provided with any mechanism of how FGF21 would cause these transcriptional changes (via PGC1a?) and whether any of these changes indeed translates into increased lipolytic or lipogenic activity.

- The authors performed bioinformatic analyses to uncover the mechanisms of FGF21-controlled browning and provide a plausible mechanism, namely PPAR α and PGC1 α , that regulate for example CPT1b expression. I understand that, due to time and costs for mouse breeding, it would be not possible to perform actual lipolysis or lipogenesis experiments on knock out tissues to verify the in silico results of this study.

- A prove for increased lipogenesis being responsible for energy expenditure in UCP-1 deficient animals is given by the decrease of OCR in the presence of Triacsin C. However, without showing that this effect is reversed in the double knock out adipose tissue, we don't know whether it depends on FGF21. Please provide the data including the double ko.

-Authors now omitted the in vitro results from the manuscript.

- If futile cycling of lipolysis/lipid synthesis is increased, there should be remodeling of iWAT towards a multilocular phenotype and the appearance of micro-lipid droplets within adipocytes. It would be desirable to include histological images showing the morphology of wt, UCP-1 ko, FGF21-ko and double-ko iWAT.

- The Authors now included histological images.

- All genotypes where raised at 30°C and then switched to 23°C and on HFD. First analyses, (including determination of FGF21 concentration in plasma) were performed 3 weeks after the switch, when body weight curves start to separate. Between week 0 and week 3 all genotypes gain similar weight. Does that indicate that it takes 3 weeks until FGF21 is increased in plasma; or that adipose tissue needs 3 weeks to be remodeled to increase energy expenditure? To understand this it would be good to show FGF21 plasma concentrations before putting the mice to 23°C and HFD, and compare it to the concentrations after 2 days, 1 week, 3 weeks, and 12 weeks at 23°C and HFD. Moreover, it should be shown whether FGF21 serum concentrations coincide with adipose tissue remodeling.

- From their response I understand that it takes at least a week to see an FGF-21 mediated effects on browning. Moreover, authors discuss that it takes 2 weeks till FGF21 increases at room temperature. Hence it makes sense to start at 3 weeks after temperature switch.

- FGF21 is known to reduce plasma TG levels (Schlein et al., Cell metab, 2016) and to affect adipocyte lipolysis. Is there any difference in plasma TG, fatty acid, or glycerol levels in the plasma of single and double ko mice on HFD and 23°C?

- Authors now included plasma concentrations of TG and NEFA. TG concentrations were lower in ucp-1 and dKO after 3 weeks of HFD compared to wt and FGF21ko. Interestingly after 12 weeks HFD FGF21 and dKO showed increased TG content compared to wt and ucp-1 ko animals. This would fit to the hypothesis that FGF21ko reverses the ucp-1 phenotype. However, TG content in the serum is lower after 12 weeks HFD compared to 3 weeks HFD feeding, which is hard to explain. Nevertheless, in the results section authors may want to indicate whether TG were increased or decreased instead of writing: “changing triglyceride concentrations”.

- FGF21 increases glucose uptake (Kharitonov et al. JCI 2005) and insulin sensitivity. Indeed, glucose and insulin concentrations are reduced in UCP-1 ko mice. However, they see no differences in glucose tolerance between the genotypes; My suggestion would be to perform ITT as it is the preferred method to measure insulin sensitivity.

- I understand that, due to time and costs for mouse breeding it is not possible to generate another cohort for ITT. Hence, HOMA-IR is fine.

- Beta klotho has been shown to be essential for FGF21 actions, and obesity is supposed to be an FGF21 resistant state with reduced Beta klotho expression (Fisher et al., Diabetes, 2010). Is there a difference in Beta klotho expression between wild-type and UCP1 deficient animals under HFD and mild cold conditions compared to 30°C chow?

- Thanks to the authors for sharing their data on beta klotho expression in iWAT of WT and UCP1 ko mice. Indeed after 12 weeks HFD feeding UCP1 ko mice show significantly increased KLB expression.

- Some methods are missing in the method section like how bomb calorimetry was performed on feces samples or how food assimilation was calculated.

- The authors included the missing methods in the manuscript.

Minor points:

- It would be better for the reader to always show energy content using the same unit, (best would be kJ/d) and not switch between kJ and kcal (Metabolic rate figure 3 a is given in kcal/h and urine energy or fecal energy, fig 4 g and fig 5 m, is given in kJ/d).

- The authors changed to unit for metabolic rate to kJ/d

- FGF21 is induced by ketogenic diet (Badman et al., Cell metab, 2007). Is its expression also induced by HFD feeding per se or only by the transfer of mice from 30°C to 23°C?

- Authors now included data showing that FGF21 is only induced when UCP1 ko mice on HFD are treated with mild cold exposure, and not by HFD per se.
- Figure 2, body weights and body fat content of the different genotypes: Is there any explanation why mice at 30°C are less obese compared to mice at 23°C? From what is known, metabolic rate is decreased at 30°C, which would lead to fat accumulation.
- The authors explanation for this discrepancy is acceptable.

Reviewer #3 (Remarks to the Author):

The authors have carefully responded to questions and criticisms. New data and changes made by the authors upon referee's suggestions have significantly improved the interpretation and discussion of the data. I consider it as an interesting contribution to this field of study.

Only two minor points to be amended: in Suppl S4 (panel A should be indicated) and legends to Suppl Figs S3 and S4 (refer to Fig 5 and not 6).

RESPONSES TO THE REVIEWERS' COMMENTS:

We are grateful to the constructive and critical comments and suggestions of all reviewers which improved the interpretation and conclusions of the findings.

Reviewer #1 (Remarks to the Author):

The revised manuscript by Keipert et al. includes several improvements relative to the first version. The way data are presented and inclusion of more balanced statements and Discussion improved the manuscript. Some of my general concerns, however remain. Regarding point 1, concerning the specificity of the effects: the authors provide, for their double KO models: a) negative data on expression (mRNA levels, not actual circulating levels) of potential brown adipokines other than FGF21, b) negative transcriptomics data on main effects in liver and skeletal muscle. These data support to some extent the specificity of FGF21 but are not fully conclusive, as stated appropriately in the new paragraph in the Discussion stating this limitation. In my opinion, however, the main remaining weakness remaining in the revised manuscript concerns the absence of a functional explanation on how the browning of WAT, in the absence of UCP1-mediated energy expenditure mechanisms, may account for protection against obesity remain (Point 3 in the previous report). This is a key and relevant point, because general assumptions in the field are that WAT browning increase energy expenditure because: a) beige adipocytes possess UCP1-dependent energy expenditure, and/or b) additional mechanisms other than UCP1 which account for energy expenditure in beige adipocytes (Spiegelman group contributions, a much less consolidated view, or other possibilities). The observations in this manuscript are important because are strongly supportive of the relevance of b), but the lack of direct assessment of what is going on in WAT in bioenergetic terms is a weakness. In their approaches at this point (and other sections of the manuscript), the authors explore these issues mostly through omics (mostly transcriptomics) data which provide steady-state data that suggest the involvement of pathways and physiological processes, but do not involve an actual exploration of function which would require direct, bioenergetically meaningful, experimentation. Moreover, the very moderate effects found when energy balance data are processed remain puzzling. Moving the data to discussion may be appropriate, but clear conclusions on whole body energy balance are nuclear to a study focused to mechanisms of obesity protection. In summary, the authors provide in their study a relevant information regarding the identification of what leads to protection against obesity in the UCP1-KO mouse model: the induction of FGF21 and resulting browning of WAT, but the mystery remains on how WAT browning in the absence of UCP1 may account for obesity protection. Without underestimating the very valuable information provided by the current manuscript, it's a pity this remains unclear, as this is a very relevant biological point.

RESPONSE: We thank this reviewer for the appreciation of our manuscript and findings, but also for his/her insightful and critical assessment that will assist new follow-up studies. We agree that the positive and negative results of this study will form a solid fundament to explore further mechanistic aspects.

Reviewer #2 (Remarks to the Author):

In their study Keipert et al., show that FGF21 is responsible for the obesity resistant phenotype of UCP1 deficient animals at ambient temperatures.

Overall, Keipert et al., answered the major and minor concerns I had, to my full satisfaction.

RESPONSE: The major and minor concerns of this reviewer were of paramount importance to improve our study and we are grateful for the independent, external view on our results and study design. We are happy that previously raised concerns could be resolved and included in the revised manuscript version.

In detail:

Major points:

- Authors nicely show that the gene expression pattern in iWAT dramatically changes when FGF21 is deficient in Ucp-1-ko animals. However, and unfortunately, we are not provided with any mechanism of how FGF21 would cause these transcriptional changes (via PGC1a?) and whether any of these changes indeed translates into increased lipolytic or lipogenic activity.
- The authors performed bioinformatic analyses to uncover the mechanisms of FGF21-controlled browning and provide a plausible mechanism, namely PPARα and PGC1α, that regulate for example CPT1b expression. I understand that, due to time and costs for mouse breeding, it would be not possible to perform actual lipolysis or lipogenesis experiments on knock out tissues to verify the in silico results of this study.

RESPONSE: We thank you for this understanding.

- A prove for increased lipogenesis being responsible for energy expenditure in UCP-1 deficient animals is given by the decrease of OCR in the presence of Triacsin C. However, without showing that this effect is reversed in the double knock out adipose tissue, we don't know whether it depends on FGF21. Please provide the data including the double ko.

-Authors now omitted the in vitro results from the manuscript.

- If futile cycling of lipolysis/lipid synthesis is increased, there should be remodeling of iWAT towards a multilocular phenotype and the appearance of micro-lipid droplets within adipocytes. It would be desirable to include histological images showing the morphology of wt, UCP-1 ko, FGF21-ko and double-ko iWAT.

- The Authors now included histological images.

- All genotypes were raised at 30°C and then switched to 23°C and on HFD. First analyses, (including determination of FGF21 concentration in plasma) were performed 3 weeks after the switch, when body weight curves start to separate. Between week 0 and week 3 all genotypes gain similar weight. Does that indicate that it takes 3 weeks until FGF21 is increased in plasma; or that adipose tissue needs 3 weeks to be remodeled to increase energy expenditure? To understand this it would be good to show FGF21 plasma concentrations before putting the mice to 23°C and HFD, and compare it to the concentrations after 2 days, 1 week, 3 weeks, and 12 weeks at 23°C and HFD. Moreover, it should be shown whether FGF21 serum concentrations coincide with adipose tissue remodeling.

- From their response I understand that it takes at least a week to see an FGF-21 mediated effects on browning. Moreover, authors discuss that it takes 2 weeks till FGF21 increases at room temperature. Hence it makes sense to start at 3 weeks after temperature switch.

RESPONSE: Thank you; these were exactly our observations and rationales for planning the experiments.

- FGF21 is known to reduce plasma TG levels (Schlein et al., Cell metab, 2016) and to affect adipocyte lipolysis. Is there any difference in plasma TG, fatty acid, or glycerol levels in the plasma of single and double ko mice on HFD and 23°C?

- Authors now included plasma concentrations of TG and NEFA. TG concentrations were lower in ucp-1 and dKO after 3 weeks of HFD compared to wt and FGF21ko. Interestingly after 12 weeks HFD FGF21 and dKO showed increased TG content compared to wt and ucp-1 ko animals. This would fit to the hypothesis that FGF21ko reverses the ucp-1 phenotype. However, TG content in the serum is lower after 12 weeks HFD compared to 3 weeks HFD feeding, which is hard to explain. Nevertheless, in the results section authors may want to indicate whether TG were increased or decreased instead of writing: "changing triglyceride concentrations".

RESPONSE: Reviewer's point well taken. We changed this in the results section.

- FGF21 increases glucose uptake (Kharitonov et al. JCI 2005) and insulin sensitivity. Indeed, glucose and insulin concentrations are reduced in UCP-1 ko mice. However, they see no differences in glucose tolerance between the genotypes; My suggestion would be to perform ITT as it is the preferred method to measure insulin sensitivity.

- I understand that, due to time and costs for mouse breeding it is not possible to generate another cohort for ITT. Hence, HOMA-IR is fine.

RESPONSE: Thank you.

- Beta klotho has been shown to be essential for FGF21 actions, and obesity is

supposed to be an FGF21 resistant state with reduced Beta klotho expression (Fisher et al., Diabetes, 2010). Is there a difference in Beta klotho expression between wild-type and UCP1 deficient animals under HFD and mild cold conditions compared to 30°C chow?

- Thanks to the authors for sharing their data on beta klotho expression in iWAT of WT and UCP1 ko mice. Indeed after 12 weeks HFD feeding UCP1 ko mice show significantly increased KLB expression.

- Some methods are missing in the method section like how bomb calorimetry was performed on feces samples or how food assimilation was calculated.

- The authors included the missing methods in the manuscript.

Minor points:

- It would be better for the reader to always show energy content using the same unit, (best would be kJ/d) and not switch between kJ and kcal (Metabolic rate figure 3 a is given in kcal/h and urine energy or fecal energy, fig 4 g and fig 5 m, is given in kJ/d).

- The authors changed to unit for metabolic rate to kJ/d

- FGF21 is induced by ketogenic diet (Badman et al., Cell metab, 2007). Is its expression also induced by HFD feeding per se or only by the transfer of mice from 30°C to 23°C?

- Authors now included data showing that FGF21 is only induced when UCP1 ko mice on HFD are treated with mild cold exposure, and not by HFD per se.

- Figure 2, body weights and body fat content of the different genotypes: Is there any explanation why mice at 30°C are less obese compared to mice at 23°C? From what is known, metabolic rate is decreased at 30°C, which would lead to fat accumulation.

- The authors explanation for this discrepancy is acceptable.

Reviewer #3 (Remarks to the Author):

The authors have carefully responded to questions and criticisms. New data and changes made by the authors upon referee's suggestions have significantly improved the interpretation and discussion of the data. I consider it as an interesting contribution to this field of study.

Only two minor points to be amended: in Suppl S4 (panel A should be indicated) and legends to Suppl Figs S3 and S4 (refer to Fig 5 and not 6).

RESPONSE: We are grateful to the reviewer to point out these mistakes and have amended these errors.